# PROST: quantitative identification of spatially variable genes and domain detection in spatial transcriptomics

Yuchen Liang [1], Guowei Shi[2], Runlin Cai[1], Yuchen Yuan[2], Ziying Xie[2], Long Yu[1], Yingjian Huang[1], Qian Shi[1], Lizhe Wang[3], Jun Li [3] ✉ & Zhonghui Tang [2] ✉

Computational methods have been proposed to leverage spatially resolved transcriptomic data, pinpointing genes with spatial expression patterns and delineating tissue domains. However, existing approaches fall short in uniformly quantifying spatially variable genes (SVGs). Moreover, from a methodological viewpoint, while SVGs are naturally associated with depicting spatial domains, they are technically dissociated in most methods. Here, we present a framework (PROST) for the quantitative recognition of spatial transcriptomic patterns, consisting of (i) quantitatively characterizing spatial variations in gene expression patterns through the PROST Index; and (ii) unsupervised clustering of spatial domains via a self-attention mechanism. We demonstrate that PROST performs superior SVG identification and domain segmentation with various spatial resolutions, from multicellular to cellular levels. Importantly, PROST Index can be applied to prioritize spatial expression variations, facilitating the exploration of biological insights. Together, our study provides a flexible and robust framework for analyzing diverse spatial transcriptomic data.

Exploring spatiotemporal patterns of transcriptional expressions in complex tissues is critical for understanding biological or pathological mechanisms[1]. Recent advances in spatial transcriptomics (ST) provide new avenues to analyze gene expression with spatial information in tissues[2]. Various ST methods have been developed, which are mainly categorized into imaging-based and next-generation sequencing (NGS)-based[3] methodologies, respectively. Imaging-based methods, including MERFISH[4], SeqFISH[5], and HybISS[6], capture the transcriptome at single-molecule resolution, while they are limited by either the small number of target genes imaged or confined detection sensitivity in imaging. NGS-based approaches, such as 10x Visium and Slide-seq[7], profile the transcriptome with increased sensitivity and throughput, while their resolutions depend on the diameter of spatially barcoded spots on microarray slides. Recently, additional NGS-based methods have

been developed with nanoscale resolution, such as DBiT-seq[8], Stereo-seq[9], and PIXEL-seq[10].

A critical analytic task in spatial transcriptomic studies is identifying spatially variable genes (SVGs) that display spatial expression patterns across spatial locations. Many methods have been proposed to elucidate the spatial variation in gene expression. Foremost of them are based on statistical models, such as trendsceek[11], SpatialDE[12], and SPARK[13]. They first construct a statistical model of the correlation between the gene expression profile and the spatial location and then return a *p*-value to indicate the spatial variability in gene expression. scGCO[14] employs a hidden Markov random field (HMRF) to assess spatial dependence under the assumption of complete spatial randomness. However, it is noticeable that parameter derivation strategies employed in those methods might pose problems, such as being sensitive to prior assumption errors and affected by local optimal

[1]School of Geography and Planning, Sun Yat-sen University, Guangzhou 510275, China. [2]Zhongshan School of Medicine, Sun Yat-sen University, Guangzhou 510080, China. [3]School of Computer Science, China University of Geosciences, Wuhan 430078, China. ✉e-mail: lijuncug@cug.edu.cn; tangzhh99@mail.sysu.edu.cn

solutions. Except for the technical challenges, the most important is that statistical modeling-based methods only use statistical significance to measure the spatial patterns in gene expressions, leading to difficulties in explaining the spatial heterogeneity and homogeneity within SVGs from a spatial perspective. Recently, methods have applied deep learning-based techniques to identify SVGs, such as SpaSEG[15] and SPADE[16]. In those methods, a neural network is constructed to integrate gene expression, spatial locations, and histological images for dimensionality reduction. Then, the low-dimensional latent embeddings are utilized for subsequent statistical models to identify SVGs. Those methods are limited by their lack of interpretability for the identified SVGs. In addition, sometimes the classic spatial autocorrelation statistics such as Moran's $I$[17] and Geary's $C$[18], are directly adopted to identify SVGs. On this basis, SINFONIA[19] ensembles these two spatial autocorrelation statistics to identify SVGs. However, Moran's $I$ and Geary's $C$ tests are insufficient to provide meaningful biological results, as they only utilize the local information of spot neighborhoods, ignoring the tissue structure in a broader range. Thus, although diverse methods have been developed for SVG detection, there is still a lack of unified criteria for quantitative evaluation of SVGs.

Another essential analytic task in ST studies is detecting spatial domains with coherent gene expressions. Clustering algorithms are often used to solve related problems[20–22] in this task. Traditional algorithms such as $k$-means and louvain[23] are directly used to cluster ST data for spatial domain identification. However, due to the challenges brought by ST data with high dimensionality and sparsity, these non-spatial methods are unfit for effectively identifying spatial domains with biological consistency. To this end, many clustering methods that focus on the specific features of ST data are currently being designed for detecting spatial domains, for instances, stLearn[24] achieving spatial smoothness under physical and morphological assumptions, and SpaGCN[25] extracting features via convolutional graph networks. Besides, there are methods that adopt a Potts model to encourage adjacent cells to share similarities in the spatial domain assignment. For example, Zhu et al. proposed an HMRF model[26] that utilizes the information from spatial adjacency relations and identifies coherent spatial domains. BayesSpace[27] and BASS[28] integrate spatial locations and gene expression into a Bayesian framework for spatial domain clustering with different sampling algorithms. In addition, other methods construct deep auto-encoder networks with different structures to learn the deep features of gene expression, such as SEDR[29], SCAN-IT[30], CCST[31], STAGATE[32], and SpaceFlow[33]. The methods reviewed above usually achieve significant improvements in the task of modeling the spatial domain at multicellular resolution from tumor and brain tissue sections, in which tissues primarily represent distinct histological architectures with reduced transcriptional heterogeneity. However, regarding the emergence of ST techniques at cellular resolution, it is still challenging to detect spatial domains at fine-resolution of complex tissues, such as from organogenesis or embryo development, encountering dynamic heterogeneity in transcription between neighborhoods within the tissue and increased noise and sparsity in measuring gene expression at the cellular level.

Here, we develop a flexible framework to quantify spatial gene expression patterns and detect spatial domains using spatially resolved transcriptomics data with various resolutions. The framework, named Pattern Recognition Of Spatial Transcriptomics (PROST), consists of two modules, PROST Index (PI) and PROST Neural Network (PNN). In the PI module, we newly created PIs as unified indicators without any statistical hypothesis for evaluating variations in spatial patterns of gene expressions. PIs facilitate the quantitative characterization of spatial gene expression patterns. In the PNN module, we utilize a neighborhood-based graph with a self-attention mechanism to integrate spatial and transcriptional information. To better delineate neighboring similarity at various resolutions, we designed an interacting process between optimizing neural network parameters and denoising low-dimensional embeddings to adaptively learn spatial dependency for achieving better accurate tissue segmentation in an unsupervised manner. The conjunction of PI and PNN modules by nature, PROST is capable to maximize the integration of spatial information and gene expression profiles to detect spatial domains. We extensively compared the performance of PROST with existing methods on ST datasets generated by different platforms (e.g., 10× Visium, Stereo-seq, Slide-seq, osmFISH and SeqFISH) with various resolutions, demonstrating that PROST exhibits superior performance in both SVG identification and domain segmentation. Notably, using cellular resolution ST data, we illustrated that PROST enables to reveal the spatial structures accurately at a single-cell resolution. Importantly, PROST Index enables to prioritize spatial variations in gene expression patterns, facilitating explorations of biological insights.

## Results
### Overview of PROST
We describe the framework of PROST using the mouse brain 10x Visium ST data as an example. As shown in Fig. 1, PROST consists of two workflows: PROST Index (PI) and PROST Neural Network (PNN). These two workflows, which are naturally connected yet independent, are introduced below.

For the high-dimensional ST data, PI interprets each gene expression matrix as an image and processes each one individually (hereinafter referring to each gene expression matrix as an image). Initially, PI interpolates irregular spots (as defined in the Methods section) to regular grids using gene expressions in tandem with spatial locations. Following preliminary pre-processing steps (such as min-max normalization and Gaussian filtering), the workflow divides a gene grid image into several sub-regions, categorizing them as either foreground or background. Subsequently, a PI score is calculated for each gene using a novel, straightforward, and efficient calculation formula based on foreground and background, aiming to quantify spatial gene expression patterns. Briefly, the PI comprises two components: the Significance and Separability factors. The Significance factor is designed under the assumption that a gene grid image would exhibit substantial differences between the foreground and background regions, and the pixel values within a connected area should display less dispersion. This factor aims to identify regions with greater homogeneity and lesser dispersion, indicating genes with significant spatial expression. Conversely, the Separability factor is developed under the assumption that there is pronounced separability between the regions of foreground and background in a gene grid image. This factor draws inspiration from the concept of spatial stratified heterogeneity, where gene expression is homogeneous within each region but varies between regions. Together, these two factors empower the PI score to effectively quantify the spatial patterns of a given gene. Additionally, we introduced both parametric and non-parametric tests for PI-based SVG identification, controlling the false discovery rate. Based on the sorting strategy of PI scores with statistical significance, we can perform downstream analysis such as SVGs detection and feature selection for dimensionality reduction analysis.

PNN first builds a directed graph to represent the spatial relationship of all spots considering its neighbors' coordinates. In the graph, the edge weight between two spots is calculated by their local neighborhood relationships, which is determined by the Euclidean distance. Next, PNN adopts a stacked low-pass denoising graph Laplacian smoothing filter to aggregate neighbor information. The smoothed feature is then fed into a self-attention mechanism to generate a meaningful low-dimensional representation. This representation integrates both spatial information and gene expressions through unsupervised learning, assigning each spot to a cluster by iteratively optimizing an objective function. In the clustering results, each cluster is considered as a spatial tissue domain. Besides, due to the integration of spatial information and gene expressions, the output

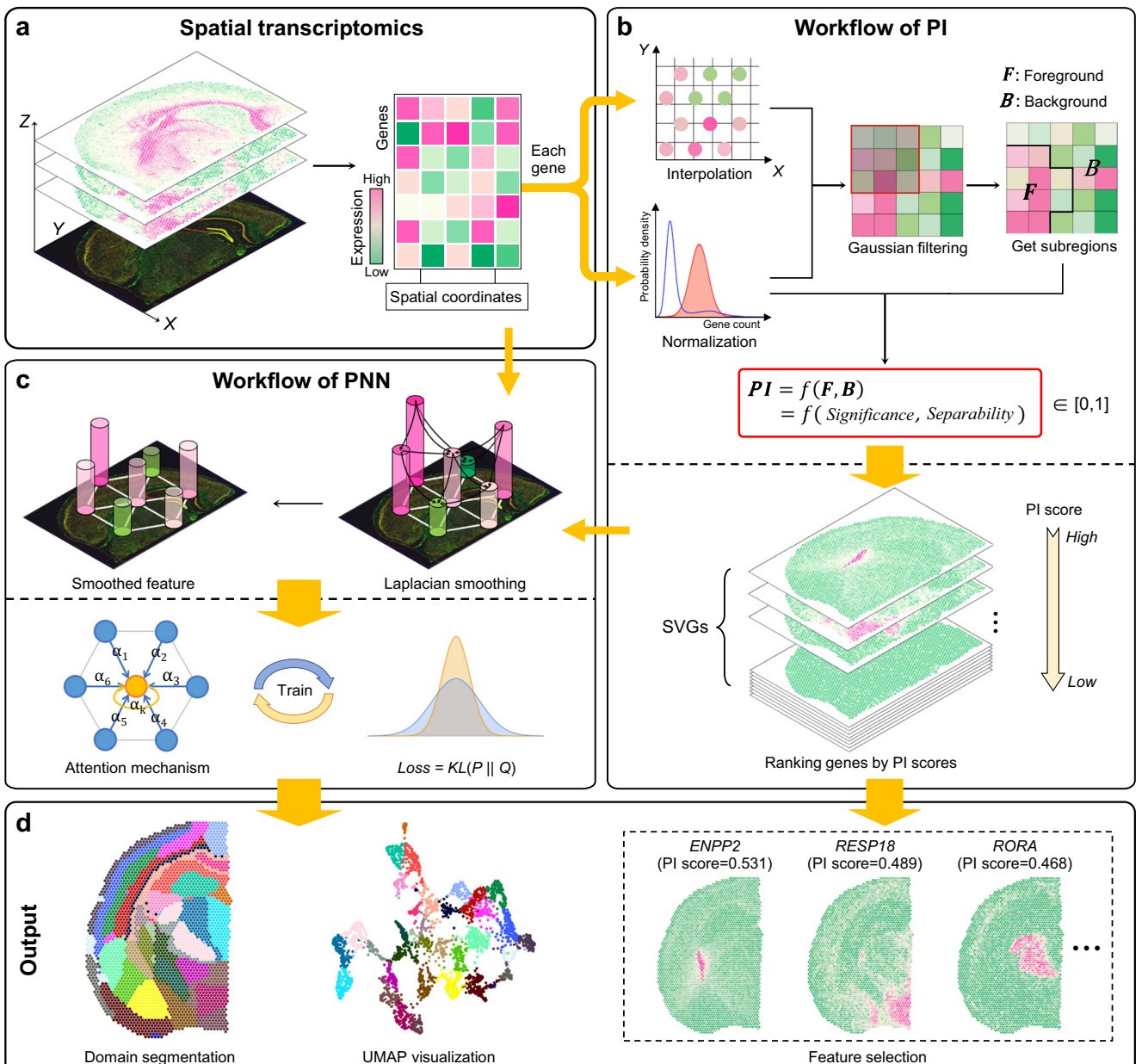

**Fig. 1 | Overview of PROST. a** Spatial transcriptomics (ST) techniques enable the simultaneous profiling of RNA transcripts with spatial locations. **b** Workflow of the calculation of PROST Index (PI). In the workflow, the spatial expression matrix of each gene is converted to an image for downstream processing. Each gene-based image is subjected to a Gaussian filtering process to suppress noise and is then divided into foreground and background, respectively, via threshold segmentation. PI scores are calculated for each gene using foreground and background signals and then applied for quantifying gene spatial expression patterns, identifying spatially variable genes (SVGs), and selecting gene features for dimensionality reduction processing. **c** Workflow of PROST Neural Network (PNN) processing. PNN is constructed using a directed graph based on spatial location pre-defined neighborhoods. PNN further uses a stacked graph Laplacian smoothing filter to aggregate neighbor information. The smoothed feature is then fed into a self-attention mechanism to generate a meaningful low-dimensional representation that integrates spatial and transcriptional information through adaptive learning. **d** The downstream analysis uses the low-dimensional representation for UMAP visualization, domain segmentation, and feature selection.

low-dimensional representation could be used for visualization in a UMAP plot[34], and to further infer the trajectory by PAGA[35].

## PROST enhances the performance of domain segmentation through learning meaningful representations

To evaluate the spatial pattern recognition performance of PROST, we compared PROST with eight existing unsupervised clustering approaches: one non-spatial method SCANPY[36] and seven spatial methods: stLearn[24], HMRF[26], BayesSpace[27], SpaGCN[25], SpaceFlow[33], STAGATE[32], and BASS[28]. We applied them to the human dorsolateral prefrontal cortex (DLPFC) 10x Visium ST dataset[37], which was manually

annotated as the cortical layers and white matter. The manual annotations were used as the ground truth to evaluate the accuracy of spatial domain identification.

To quantitatively compare the performance of domain segmentation, we employed the Adjusted Rand Index (ARI)[38] and Normalized Mutual Information (NMI)[39] metrics to measure the similarity between the predicted domains and the manual annotations across all twelve sections of the DLPFC dataset. Notably, PROST presented more consistent ARI scores across the twelve DLPFC datasets than other benchmark methods did (Fig. 2a), highlighting the robustness of PROST in domain segmentation. Moreover, PROST significantly

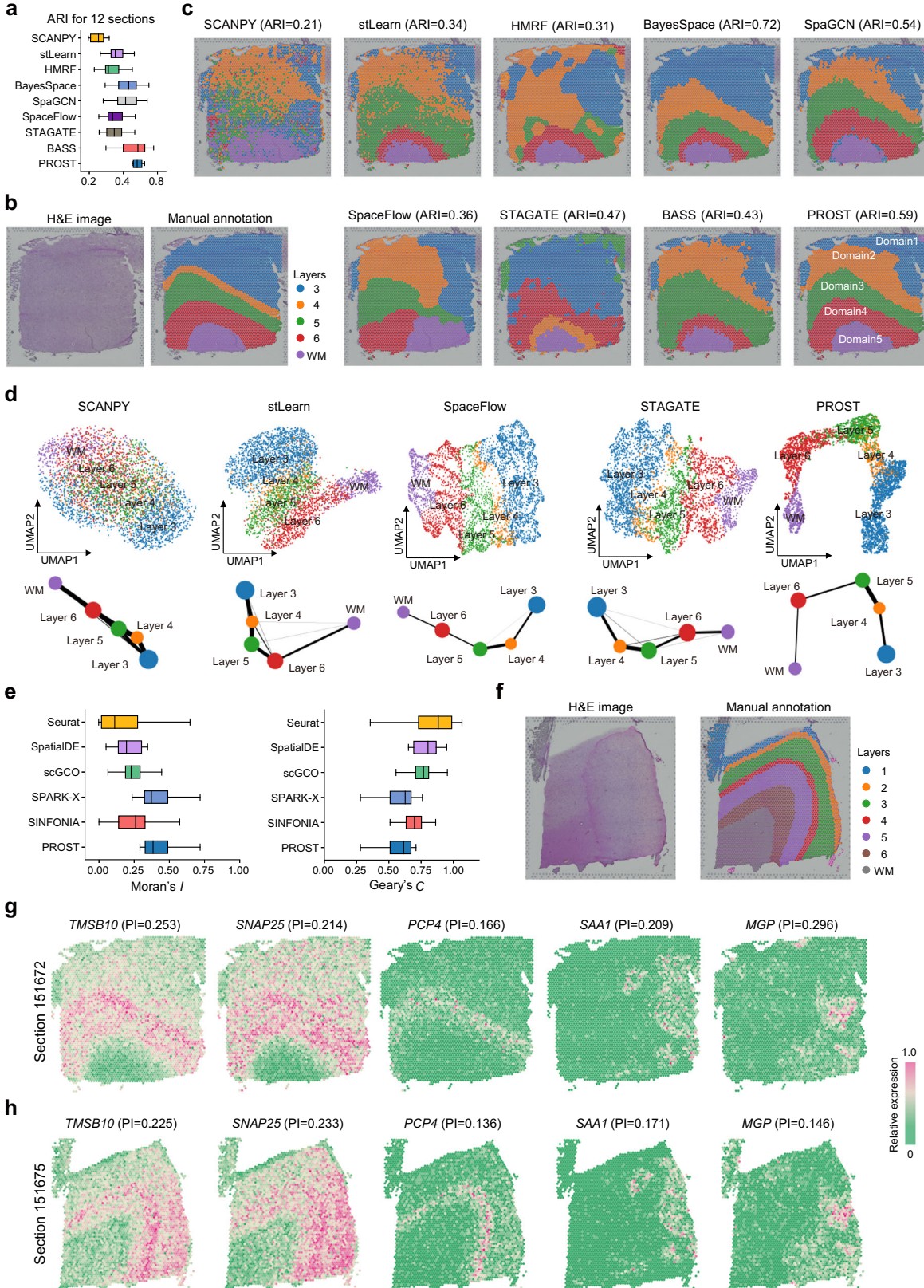

elevated the ARI scores in comparison to other benchmark methods (Kolmogorov–Smirnov test, $p$ value < 0.05), with the exceptions of BASS and BayesSpace. Upon a more detailed comparison, PROST emerged with both the highest average ARI score (0.474) and average NMI score (0.610), showcasing superior performance compared to both BASS and BayesSpace (Fig. 2a, Supplementary Tables 3 and 4).

Collectively, these findings indicate that PROST has the best capacity in identifying the spatial domain structure in the DLPFC dataset.

Specifically, given section 151672 of the DLPFC dataset as an example, the non-spatial method, SCANPY, struggled to clearly identify the layer structure or distinct patterns of the tissue section compared with the spatial methods (Fig. 2b, c). Although almost all spatial

**Fig. 2 | PROST improves the identification of spatial domains and the detection of SVGs in the human dorsolateral prefrontal cortex (DLPFC) tissue. a** Boxplot shows the Adjusted Rand Index (ARI) to summarize the domain segmentation accuracy of each method in all 12 sections of the DLPFC dataset. The boxplot's center line, box limits, and whiskers denote the median, upper and lower quartiles, and 1.5× interquartile range, respectively. Source data are provided as a Source Data file. **b** Hematoxylin and Eosin (H&E) image and the manual annotation of the DLPFC section 151672. **c** Spatial domains identified by SCANPY, stLearn, HMRF, Bayes-Space, SpaGCN, SpaceFlow, STAGATE, BASS, and PROST, respectively, in the DLPFC section 151672. **d** UMAP visualizations (top) and PAGA graphs (bottom)

generated by SCANPY, stLearn, SpaceFlow, STAGATE, and PROST, respectively, for the DLPFC section 151672 dataset. The UMAPs and PAGA graphs were colored by the corresponding layer annotation of spots in (**b**). **e** Boxplot shows the Moran's $I$ and Geary's $C$ values for spatial autocorrelation using 50 top-ranked SVGs detected by Seurat, SpatialDE, scGCO, SPARK-X, SINFONIA and PROST, respectively, in the DLPFC section 151672 dataset. The boxplot legend is the same as (**a**). Source data are provided as a Source Data file. **f** H&E image (left) and the manual annotation (right) of the DLPFC section 151675. Representative SVGs detected by PROST in the DLPFC section 151672 (**g**) show the same spatial patterns in the DLPFC section 151675 (**h**), demonstrating the transportability of SVGs.

methods could detect five visible domains in the tissue section, there remain some deficiencies of existing spatial approaches, such as the difficulty in recognizing the accurate and clear boundaries between domains to be consistent with the manual annotation (Fig. 2b, c). For example, HMRF inappropriately merged Layers 4 and 5 together. SpaceFlow could recognize spatial domains with obvious boundaries but struggled to accurately determine the genuine spatial relative position of layers as marked in the manual annotation. BASS inadvertently grouped Layers 5 and 6 into one category. In contrast, PROST accurately identified the structure of white matter and nearby cortical layers, matching closely with the manual annotation (Fig. 2b, c). PROST also delineated clear boundaries between layers (Fig. 2b, c), although its performance (ARI = 0.59) is lower than that of BayesSpace, which exhibited the highest ARI score of 0.72. The high ARI score of Bayes-Space resulted from its domination detection of Layer 3, wherein PROST separated it into two domains (Fig. 2c). Notably, although this region was manually annotated as one layer, it is likely to encompass more than one domain in transcriptional features[33]. In line with this, SpaGCN, stLearn, and STAGATE also detected two unexpected domains in the annotated Layer 3 (Fig. 2c). However, their ARI scores were lower than that of PROST, indicating PROST outperformed competing methods with its superior sensitivity in detecting transcriptional heterogeneity that was not evident in histological annotation.

Next, the embeddings generated by all compared methods were applied to the uniform manifold approximation and projection (UMAP) space for visualization and to PAGA for inferring trajectory, respectively. In the UMAP space with PROST embeddings, spots belonging to the same manually annotated layer clustered together, while spots from distinct layers barely commingled (Fig. 2d), indicating PROST effectively reveals the layered spatial organization of the mouse brain tissue accurately. Furthermore, the PAGA graph with PROST embeddings showed an apparent developmental path of cortical layers (Fig. 2d), consistent with the chronological development order of cortical layers. In comparison, spots from different annotated layers intimately mingled together when using the embeddings generated by SCANPY and stLearn in the UMAP space (Fig. 2d). Similarly, SCANPY and stLearn embeddings also resulted in indistinct trajectories inferred by PAGA (Fig. 2d). For SpaceFlow and STAGATE, low-dimensional embeddings showed apparent separations in the UMAP space but failed to maintain the features of local tissue structures (Fig. 2d), representing the spots from the same layer were not closely cohesive together. Consistently, the PAGA trajectories constructed using SpaceFlow and STAGATE embeddings displayed only moderately clear developmental paths. HMRF, BayesSpace, SpaGCN, and BASS did not provide the low-dimensional embedding and were therefore excluded from the above comparisons.

Further analysis revealed that PROST enhanced domain segmentation performance across the additional sections of the DLPFC dataset as well as in datasets derived from tissues with varied complexity and generated using different platforms with distinct resolutions (Supplementary Figs. 1–21).

We acknowledged that the performance of methods can vary significantly across datasets due to differences in data complexity

and the features each method is designed to capture. Despite STA-GATE, SpaceFlow, and other methods not being the top performers on the DLFPC dataset, they may exhibit superior performance on other datasets. Therefore, in our subsequent comparisons, we have retained these methods to further assess and compare them with PROST.

Together, these comparisons indicate that PROST effectively retains transcriptional and spatial information across diverse ST datasets, thus generating spatially and chronologically consistent low-dimensional representations for accurate tissue segmentations.

## PROST quantitatively identifies SVGs
We evaluated the performance of PROST in identifying SVGs, comparing with four representative methods: Seurat[40], SpatialDE[12], SPARK-X[41], scGCO[14], and SINFONIA[19]. To assess the credibility of SVGs detected by these methods, we used the spatial information of SVGs to calculate the Moran's $I$[7] and Geary's $C$[18] metrics, respectively.

In section 151672 of the DLPFC dataset, as analyzed above, PROST identified 4,684 SVGs, which passed both parametric and nonparametric tests (FDR < 0.05) with PI scores greater than 0 (Supplementary Fig. 22a). SpatialDE and SPARK-X detected 1346 and 5721 SVGs (adjusted $p$ value < 0.01), respectively (Supplementary Fig. 22a). Both Seurat and SINFONIA were configured to detect 3000 SVGs for the downstream comparison. Notably, among the methods considered, SpatialDE detected the fewest SVGs. Furthermore, of the SVGs detected by SpatialDE, 208 have an adjusted $p$ value equal to 0, leading to an inability to evaluate the significance of spatial heterogeneity in gene expression between SVGs. Notably, the PI scores as indicators for spatial variation in gene expression could be applied to rank SVGs quantitatively. The SVGs detected by PROST with top ranking showed substantial spatial patterns (Supplementary Fig. 22f). There were noisy SVGs detected by SPARK-X with the most significant adjusted $p$-values (Supplementary Fig. 22g), although the number of SVGs detected by SPARK-X was slightly higher than that by PROST. The consistent significance for the SVG identification by PROST was also observed in other benchmark DLPFC datasets (Supplementary Figs. 1–11).

For further comparisons, we selected the 50 top-ranked SVGs from each method according to the sorted PI scores or adjusted $p$ values, respectively, then applied them to calculate Moran's $I$ and Geary's $C$ metrics. The results showed that the median Moran's $I$ value was 0.384 for PROST, the highest of all methods (0.372 for SPARK-X, 0.260 for SINFONIA, 0.228 for scGCO, 0.195 for SpatialDE, and 0.111 for Seurat, respectively) (Fig. 2e and Supplementary Fig. 22b). We next compared the two best methods, SPARK-X and PROST. Among the 50 top-ranked SVGs, there were 40 shared genes (Supplementary Fig. 22c). However, for the remaining 10 SVGs in each method, we found the SVGs only detected by PROST represented higher Moran's $I$ and lower Geary's $C$ values, respectively, than those by SPARK-X (Supplementary Fig. 22d), indicating that PROST possessed a better capability in identifying genuine SVGs. Next, we applied PI scores to quantify the variations between SVGs exclusively detected by each method. These results show that SVGs uniquely detected by PROST showed better metrics, compared with the SVGs only identified by SPARK-X (Supplementary Fig. 22e). These findings suggest that PI is a

robust quantitative index for identifying SVGs with strong spatial autocorrelation and distinct spatial patterns.

In addition, the SVGs detected by PROST allow for directly marking spatial domains, without any previous steps taken, such as differential expression analysis. As top-ranked SVGs, *TMSB10* (PI Score = 0.253, No. 23 in ranking, FDR = 0) and *SNAP25* (PI score = 0.214, No. 36 in ranking, FDR = 0) distinctively marked Domain 3 (Fig. 2g). The analysis of differentially expressed genes between domains further validated that *TMSB10* and *SNAP25* were significantly enriched in Domain 3 (adjusted *p* value = 4.96e-125 and 1.12e-37, respectively). Although *TMSB10* and *SNAP25* exhibited visual similarity in spatial expression patterns, the PI scores effectively indicated the subtle differences in spatial coherence between *TMSB10* and *SNAP25*, in which *TMSB10* showed a higher PI score with low spatial dispersion than *SNAP25*. Similarly, *PCP4* with a moderately high PI score (PI score = 0.166, No.69 in ranking, FDR = 0) also clearly marked Domain 3 (Fig. 2g). Our results indicate that the PI score is a strong indicator for prioritizing spatial gene expression patterns, which cannot be informed by the significance of *p*-values from the existing methods.

It is worth noticing that the PI score consists of two metrics, *Separability* and *Significance*, which interact in harmony to characterize the spatial dispersion (homogeneity or heterogeneity, *Separability*) and spatial enrichment (*Significance*) of spatial gene expressions in the tissue section. For example, the expression of *SAA1* (PI score = 0.209, No. 43 in ranking, FDR = 0) demarcated the top right part of Domain 1 (Fig. 2g), representing a much high *Separability* value (0.334), in contrast, a much low *Significance* value (−1.56e-4). The measurements of *Separability* and *Significance* indicated that *SAA1* expression displayed a substantial homogeneity in a spatial region but subtle enrichment in the expression level compared to the background. Like *SAA1*, *MGP* also possessed extremely disparate *Separability* (0.381) and *Significance* (0.016) values in spatial gene expression (Fig. 2g). These results illustrate that the *Separability* and *Significance* measurements of PI can be jointly applied to characterize the degrees of spatial dispersion and enrichment in spatial gene expression patterns.

Furthermore, given the fact that the same SVG should share similar spatial expression patterns across different sections from the same tissue regions of distinct subjects, we hypothesized that genuine SVGs detected by PROST should be transportable between those tissue sections. To this end, we chose representative SVGs from section 151672 (as analyzed above), then applied them to a distinct section 151675 of the same tissue region but from a different subject[37], validating their transportability by visualization and comparison of PI scores. We confirmed that the SVGs exhibited highly similar spatial expression patterns in distinct tissue sections with close comparable PI scores (Fig. 2g, h). Additional examples further illustrated that the SVGs detected by PIs displayed transportability in spatial expression patterns across the twelve sections of the DLPFC datasets (Supplementary Fig. 23).

Except for the DLPFC datasets, we extended our evaluation of the robustness of PI-based SVG identifications to additional datasets, representing various tissues and experimental conditions. We observed that the majority of SVGs with PI scores greater than 0 exhibited statistical significance in both parametric and non-parametric tests across diverse datasets (Supplementary Fig. 24), underscoring the robustness of PI-based SVG identifications. Moreover, our results illustrated the resilience of PI in detecting SVGs under varying noise conditions, with superior accuracy and control of false positive rates compared to the HMRF model-based scGCO (Supplementary Fig. 25). Additionally, PI performance is comparable to the neural network-based method STAGATE under diverse noise conditions (Supplementary Fig. 26).

In summary, PI is capable of quantitatively characterizing spatial variations in gene expression patterns without any statistical hypothesis.

## PI-based SVG selection enhances the domain segmentation performance of PROST

Given our observation that the PI demonstrates robustness in SVG identification, we postulate that SVG selections guided by PI could potentially optimize the domain segmentation performance of PROST. To this end, we first compared the domain segmentation results generated by the PNN module of PROST using two distinct inputs: the entire gene set and varying numbers of SVGs selected through different methods. Our analyses revealed that the accuracy of domain segmentation achieved by PNN substantially improved when utilizing SVGs selected based on the PI, compared to the case when PNN was supplied the entire gene set as input (Supplementary Fig. 27a). Notably, the accuracy of domain segmentation improved with the increase in the number of SVGs selected using PI, achieving optimal results with 3000 SVGs (Supplementary Fig. 27a). We speculate that the optimal number of SVGs might differ across ST datasets, contingent on the inherent noise of the respective dataset. Moreover, we compared domain segmentation results produced by PNN using SVGs selected through various methods. This comparative analysis underscored the superiority of the PI in SVG selection, outperforming other methods such as Seurat, SINFONIA, and SPARK-X (Supplementary Fig. 27a).

Next, we replaced PNN with the benchmark methods for spatial domain identification, including STAGATE, SpaceFlow and SpaGCN, and compared the domain segmentation results, using SVGs selected by PIs and other different methods. Our analysis revealed that the PI can also dramatically enhance the domain segmentation performance of STAGATE and SpaceFlow (Supplementary Fig. 27b, c). However, when utilizing SpaGCN, comparable performances were achieved whether SVGs were selected based on PIs or by default, using the entire gene set as input (Supplementary Fig. 27d).

We lastly assessed the performance of the joint utilization of the PI and PNN modules in the presence of noise, particularly given the inherent limitations of current ST technologies. To evaluate this, we adopted a simulation method that combines real spatial locations with simulated gene expression data to produce semi-synthetic datasets. This method was applied to the mouse somatosensory cortex osmFISH data using scDesign3, as outlined by ref. 75. Our analysis reveals that PROST maintains a superior robustness in relation to the drop-off of sequencing depth, outperforming the benchmark methods (Supplementary Fig. 28).

Together, our findings demonstrate that PI-based SVG selection can enhance domain segmentation of PROST, even in scenarios with the intrinsic noise present in ST data.

## PROST improves the clustering of recognized layers in mouse olfactory bulb tissue at cellular resolution

To evaluate the performance of PROST on ST datasets with fine resolution, we chose an ST dataset with cellular resolution (-14 μm in diameter per spot) generated by the Stereo-seq platform from the mouse olfactory bulb tissue[9]. In the original study, the mouse olfactory bulb was annotated into nine laminar layers according to the DAPI-stained image (Fig. 3a). We directly adopted the annotation as the ground truth of the tissue structure to compare PROST with existing methods. After filtering out spots that uncovered by the tissue section, the data matrix consists of 27,106 genes across 19,109 spots.

We first compared the domain segmentation performance of SpaGCN, STAGATE, SpaceFlow, BASS, and PROST on this cellular resolution dataset. As shown in Fig. 3b, PROST recognized clear structures of the laminar organization of mouse olfactory bulb from inside to outside. Some subtle layers, such as the narrow internal plexiform layer (IPL), mitral cell layer (MCL), and glomerular layer (GL), were accurately identified by PROST. Most notably, PROST successfully revealed the sub-layers for the olfactory nerve layer (ONL, ONL_1 and ONL_2 in Fig. 3e) and the external plexiform layer (EPL, EPL_1 and EPL_2 in Fig. 3e), respectively. In contrast, STAGATE, SpaceFlow, and

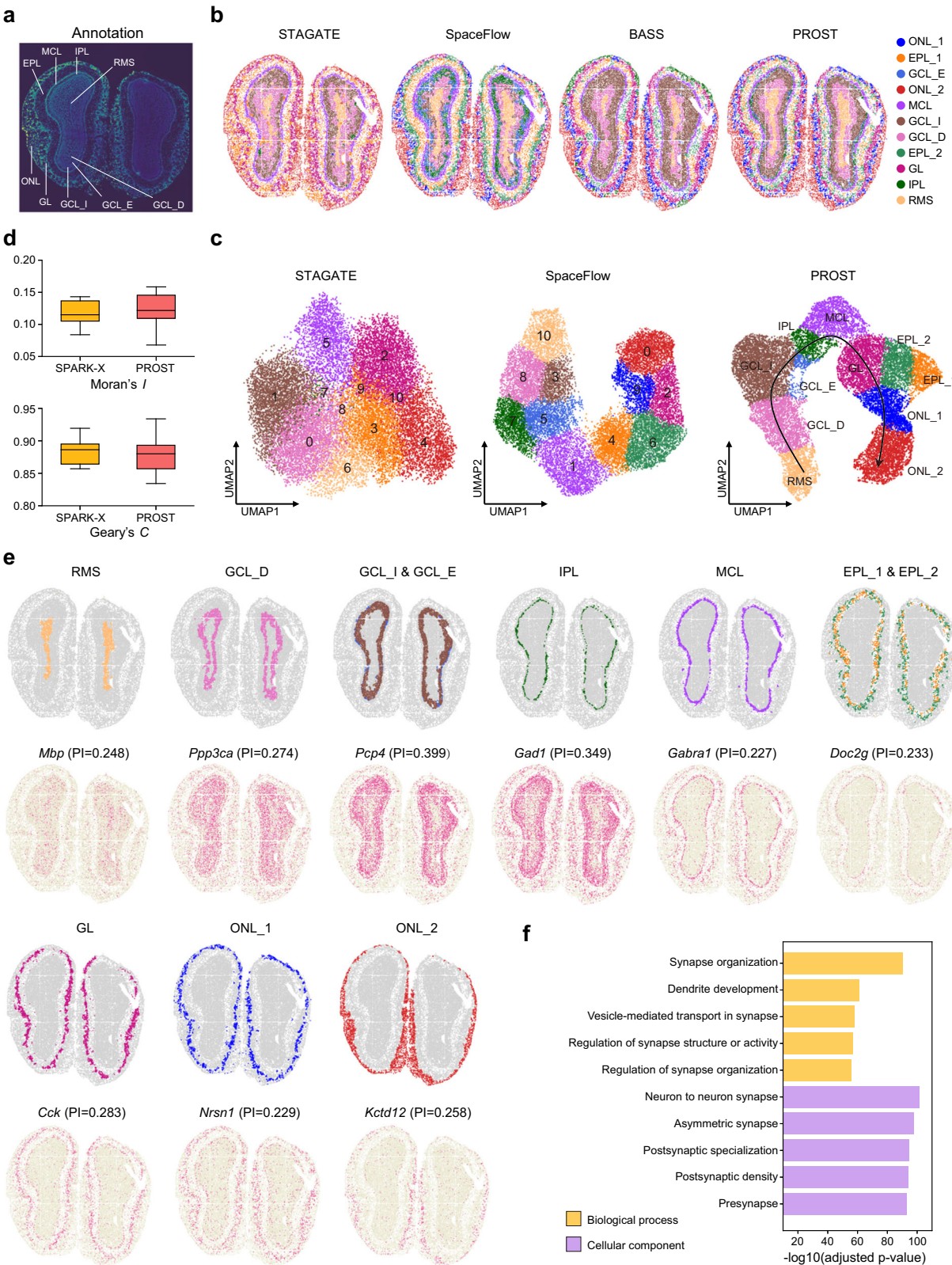

BASS were unfitted to reveal the clear tissue structures of the mouse olfactory bulb tissue, especially to distinguish the three-layer structure of the granule cell layer (GCL) (Supplementary Fig. 29). In UMAP visualization (Fig. 3c), we observed that the latent representation generated by PROST well preserved both the spatial information and gene expression profiles of the cellular resolution ST data. In addition,

the recognized tissue layers represented a clear developmental trajectory in the UMAP space, in line with the developmental sequence of these layers[42]. As a comparison, the low-dimensional embeddings generated by STAGATE were insufficient to preserve spatial information, leading to a limitation in distinguishing spatial domains. Although SpaceFlow, to some degree, preserved the global spatial information

**Fig. 3 | PROST reveals spatial cellular patterns in mouse olfactory bulb tissue using Stereo-seq data at single-cell resolution. a** Laminar organization of the mouse olfactory bulb annotated in the DAPI-stained image generated by the original paper of SEDR. **b** Spatial domains identified by STAGATE, SpaceFlow, BASS, and PROST, respectively, with a fixed number of clusters ($n = 11$) as a clustering parameter. PROST's spatial domains were annotated by marker genes, which are ONL_1: olfactory nerve layer_1, EPL_1: external plexiform layer_1, GCL_E: granule cell layer externa, ONL_2: olfactory nerve layer_2, MCL: mitral cell layer, GCL_I: granule cell layer internal, GCL_D: granule cell layer deep; EPL_2: external plexiform layer_2, GL: glomerular layer; IPL: internal plexiform layer; RMS: rostral migratory stream, from inside to outside, respectively. **c,** UMAP visualizations generated by STAGATE, SpaceFlow, and PROST, respectively. The UMAPs were colored according to the corresponding layer annotation of spots in (**b**). **d** Boxplot shows the Moran's *I* and

Geary's *C* values that were calculated for the 20 top-ranked SVGs detected by PROST and SPARK-X, respectively. The boxplot's center line, box limits, and whiskers denote the median, upper and lower quartiles, and 1.5× interquartile range, respectively. Source data are provided as a Source Data file. **e** Visualization of spatial domains (top) identified by PROST and spatial expression patterns of the corresponding marker genes (bottom). The annotation of spatial domains is the same as in (**b**) (PROST). **f** Gene Ontology (GO) enrichment analysis for the SVGs detected by PROST. The length of bars represents the enrichment of GO terms using -log10(FDR adjusted *p* value) metric from topGO analysis. Bars are colored into two categories according to biological process (yellow) and cellular component (purple). *P* values were obtained using the one-sided Fisher's exact test with FDR correction. Source data are provided as a Source Data file.

in dimensionality reduction, the local spatial information was partially dropped, resulting in a challenge in generating the spatial consistent low-dimensional embeddings.

To analyze the identified domains by PROST from a biological aspect, we analyzed domain-specifically expressed genes and found that most domain-specific genes represented exact spatial expression patterns in their corresponding domains. For instance, the mitral cell marker gene *Gabra1* was significantly expressed (adjusted *p* value = 2.07e-104) in the identified narrow tissue structure MCL (Fig. 3e). One of the top-ranked genes, *Pcp4*, showed clear expression patterns in the identified GCL-I (adjusted *p*-value = 6.91e-4) and GCL-E (adjusted *p*-value = 0), a critical determinant of synaptic plasticity in the cerebellum and locomotor learning[43]. While gene *Ppp3ca* precisely represented in the identified GCL-D (adjusted *p* value = 2.63e-110), consistent with the previous report, ~10% of GABAergic granule cells are enkephalin-like immunoreactive and mainly located in the deeper half of the GCL[44]. In addition, *Nrsn1*, which might be relevant for learning and memory[45], was primarily expressed in the additionally identified ONL_1 (adjusted *p* value = 1.65e-17). Some previously known marker genes, such as *Doc2g* and *Kctd12*[46], were also identified by PROST to indicate the segmented spatial domains. The UMAP visualization of these marker genes further confirmed that PROST possessed an excellent capability to generate spatially and chronologically consistent low-dimensional representations (Supplementary Fig. 30a, b).

Finally, we compared the performance of SVGs identification between PROST and SPARK-X. For this dataset, PROST detected 2,527 SVGs, much more than those detected by SPARK-X (1749, adjusted *p* value < 0.01), indicating PROST possessed a sensitive characteristic in SVG detection for the high-resolution ST data. We calculated Moran's *I* and Geary's *C* for the 20 top-ranked SVGs identified by PROST and SPARK-X, respectively, quantifying the spatial patterns of those top-ranked SVGs. The median Moran's *I* value was 0.122 for PROST against 0.115 for SPARK-X, and the median Geary's *C* value was 0.880 for PROST against 0.886 for SPARK-X (Fig. 3d), indicating PROST detected SVGs with a greater spatial autocorrelation. We further performed a Gene Ontology (GO) analysis for the SVGs detected by PROST (Fig. 3f). For biological process, the significantly enriched GO terms were associated with the synaptic organization (GO:0050808, adjusted *p* value = 5.05e-91), regulation of synapse organization (GO:0050807, adjusted *p* value = 2.21e-56), with a representative gene *Malat1* that is involved in synapse formation or maintenance[47], and dendrite development (GO:0016358, adjusted *p* value = 1.33e-61) with a representative gene *Ppp3ca* that is involved in long-term synaptic potentiation[48]. We also found several GO terms from cellular function significantly enriched, reflecting neuron function (Fig. 3f). In addition, almost all marker genes and GO terms-related representative genes exhibited high PI scores (Supplementary Fig. 30c), indicating that PROST is capable of identifying biologically meaningful SVGs.

Taken together, PROST outperformed competing methods at cellular resolution by accurately capturing the ground truth of domain segments and identifying genuine SVGs.

## PROST reveals insights of spatial patterns over mouse embryogenesis at cellular resolution

SeqFISH, a highly multiplexed FISH (fluorescence in situ hybridization)-based method, can simultaneously profile the expression of hundreds or thousands of genes within single cells, and their spatial locations are preserved[49]. Utilizing seqFISH to target 351 genes, Lohoff et al. recently generated a spatially resolved gene expression map across the entire embryo at single-cell resolution[50]. However, it is a tremendous challenge to elucidate the architectural aspects of the complex tissue due to the elusive boundary forming during embryogenesis, with a limited number of genes for feature extraction in low-dimensional embeddings. To this end, we applied PROST to Lohoff et al.'s dataset to evaluate its general applicability. In the original study, the embryo tissue section was annotated with 24 tissue structures (embryo 1, Fig. 4a), which were adopted for the comparison with domain segmentations generated by PROST.

Firstly, PROST successfully recognized the distinct tissue structures of the mouse embryo (embryo 1, Fig. 4b), which were largely similar to the tissue architectures annotated in the original study but differed in some subtle tissue structures. We next detected marker genes for each PROST-identified domain to validate those different subtle structures (Supplementary Fig. 31). For example, Domain 13 showed a subtle difference in structure compared to the cardiomyocytes region annotated in the original study (Fig. 4a, c). However, the cardiomyocyte marker gene, *Ttn*[51], was exclusively expressed in Domain 13 (adjusted *p* value = 0), indicating that Domain 13 is related to the development of heart tube[50]. In addition, Domain 16, defined by PROST, apparently differed from the tissue structure annotated in the original study. In consistence, we observed the mesoderm marker gene *Foxf1* primarily expressed in Domain 16 (adjusted *p* value = 0), which gene involved in celom formation, localizing at the lateral plate mesoderm and the extraembryonic mesoderm of the allantois[52]. Other domains that differed between PROST and the original study were shown in Fig. 4c, accompanied by domain-specific genes.

Most importantly, PROST precisely recognized the domains for the forebrain, midbrain, and hindbrain (embryo 1, Fig. 4d), which were grouped as an intermingled domain using the entire dataset for clustering in the original publication[50]. Consistently, PROST also accurately identified these domains for the forebrain, midbrain, and hindbrain in additional embryos (embryo 2 and 3, Supplementary Fig. 32). Our analysis demonstrated that PROST effectively dissected the histological structures within this complex region, delineating clear tissue boundaries. The segmented domains were further validated by the spatial expression patterns of marker genes. For instance, *Six3*, an anteriormost brain marker[53], was primarily expressed in Domain 5 (adjusted *p* value = 1.36e-73), demarcating the anterior-most developing brain region (prosencephalon), the forebrain's embryonic precursor. Adjacent to Domain 5, marker gene *Lhx2*, a LIM homeobox gene mainly expressed in the forebrain[54], clearly corresponded to Domain15 (adjusted *p* value = 5.46e-193). Bordering on Domain 15, *Fezf1* was expressed mainly in Domain 10 (adjusted *p* value = 2.23e-81),

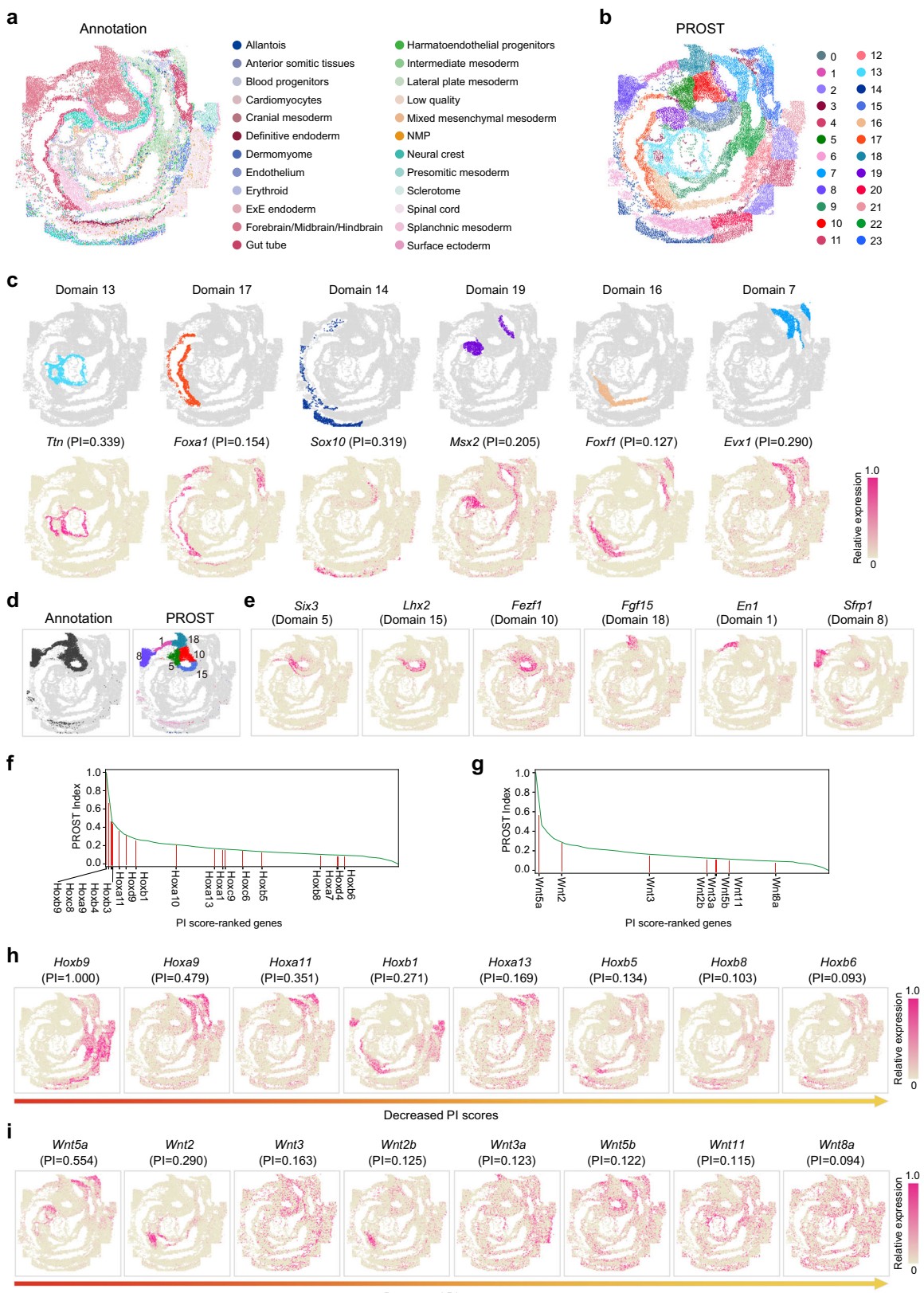

**Fig. 4 | PROST deciphers spatial cellular patterns in mouse embryogenesis using SeqFISH data at single-cell resolution. a** Annotation of SeqFISH-profiled mouse embryo tissue sections, which was obtained from the original publication. **b** Spatial domains identified by PROST using the SeqFISH data with a fixed number of clusters (*n* = 24) as a clustering parameter. **c** Visualization of representative spatial domains (top) identified by PROST and spatial expression patterns of the corresponding marker genes (bottom). **d** Spatial location of forebrain/midbrain/ hindbrain in the original annotation[50] (left) and the corresponding domains segmented by PROST (right). **e** Marker genes of the corresponding domains in forebrain/midbrain/hindbrain segmented by PROST. Distribution of PI scores of *Hox* (**f**) and *Wnt* gene families (**g**). Source data are provided as a Source Data file. Spatial expression patterns of the top-ranked members of *Hox* (**h**) and *Wnt* gene families (**i**) identified by PROST.

controlling neuronal differentiation at the forebrain region during early embryogenesis[55]. Thus, the neighboring Domains 10 and 15 delineated the spatial sub-regions of the forebrain. *Fgf15* regulates the postmitotic transition of dorsal neural progenitors, controlling dorsal midbrain neurogenesis's initiation and proper progression[56]. We found *Fgf15* substantially expressed in Domain 18 (adjusted *p* value = 3.12e-115), indicating that PROST correctly defined the midbrain region. In addition, one of the top-ranked marker genes of Domain 1, *En1* (adjusted *p* value = 1.22e-134), displays a high expression level at the midbrain-hindbrain boundary[57], indicating that Domain 1 represented the domain of the mid-hindbrain junction. Furthermore, *Sfrp1* was highly expressed in Domain 8 (adjusted *p* value = 0), indicating that Domain 8 represented the presumptive hindbrain[58]. These results demonstrate that PROST possessed the robust ability for domain segmentation in the complex tissue at single-cell resolution.

Next, we evaluated the capability of PROST for SVGs identification using the SeqFISH dataset. We observed that *Hox* gene family members were among the genes with the most variability in the spatial expression patterns (Fig. 4f, h), consistent with the "Hox code" that is mainly expressed along the anterior–posterior axis[59] in the combination of functional activities. Similarly, *Wnt* family members represented large variations in the spatial expression patterns (Fig. 4g). By ranking the PI scores, we could directly sort out the *Wnt* family members that were expressed in a spatially restricted and dynamic patterns in embryogenesis (Fig. 4i). Our analysis indicated that PROST possessed the general applicability to recognize SVGs with biological significance.

Together, PROST enables the correct clustering of spatial domains in the complex embryonic section and provides a better understanding of the molecular architecture of the developing mouse brain.

## Discussion

In this study, we presented PROST, a quantitative pattern recognition framework for spatial transcriptomic data analysis. We evaluated the performance of PROST on a diverse array of ST data generated by different platforms with various spatial resolutions from multicellular to cellular resolutions. Our results demonstrate that PROST could effectively improve domain segmentation performance and detect SVGs with much clearer spatial expression patterns and biological interpretations than previous methods. Additionally, PROST is flexible with parameter setting (Supplementary Fig. 33) and exhibits reasonable efficiency in both memory usage and computational time (Supplementary Figs. 34 and 35), with the potential scalability for the analysis of anticipated larger datasets in the future (Supplementary Fig. 36).

PROST, as a flexible framework, consists of two modules, PI and PNN, which can be applied independently or integrated into existing analysis workflows, while a more important aspect is that the joint utilization of those two workflows enables PROST to maximize the integration of spatial information and gene expression profiles to detect spatial domains in coherent expression patterns with histological characteristics. PI that we originally created in this study is the only existing index quantifying spatial gene expression patterns with the consideration of tissue context and expression profiles. PI possesses the capability for biological interpretation in prioritizing spatial gene expression variations. As a powerful indicator, we expect that PI would facilitate our understanding of the biological function of SVGs in tissue domains. More fundamentally, SVGs are naturally associated with depicting spatial domains but are technically dissociated in most existing methods. Thus, PI could be applied as a reference for feature selection in dimensionality reduction analysis and can be flexibly used to support existing methods in determining low-dimensional representation. Distinguished from the one-round clustering algorithms, the PNN module with a self-attention mechanism essentially promoted the

adaptive learning of spatial dependency in transcription information, enabling us to achieve more detailed/accurate tissue segmentation results. The flexibility of utilizing spatial information in a different range of neighborhoods provided PROST with the robust adaptation for multiple types of ST data in various resolutions. Importantly, for the ST data with a high spatial resolution, PROST is superior in recognizing subtle tissue structures compared to the existing methods.

The limitation of PROST is that the PNN workflow disregarded the tissue context information of the histological image, leading to the situation where domain segmentation is mainly determined by spatial gene expression profiles. To attenuate this limitation, we introduced Laplacian smoothing to aggregate neighboring gene expression information in the pre-processing steps, reducing the influence of abnormal gene expressions on domain segmentation. In addition, the PI score-based feature selection also alleviated the impact of lacking use of histological information for clustering tissue domains. Future improvement could be achieved by directly integrating histological images and spatial gene expression profiles for tissue domain determination. Furthermore, the PI-based quantification of spatial gene expression patterns could be applied to calculate the contribution of gene expression variation in tissue domains, enabling the discovery of the function of some gene combinations in specific tissue domains.

Lastly, given the growing volume of scRNA-seq and ST data available in public repositories, forthcoming versions of PROST might well consider integrating reference-based strategies, especially as the field is foreseeing a surge in such methodologies[60,61]. By utilizing reference atlases and harnessing transfer learning techniques, PROST has the potential to create an even more precise adjacency graph, substantially enhancing spatial domain segmentation, particularly within intricate tissue structures.

## Methods
### Data preparation
PROST requires input of gene spatial expression data **X**, which is stored in an $N \times D$ unique molecular identifier (UMI) count matrix with $N$ spots and $D$ genes, including the spatial location information of each spot in the form of two-dimensional coordinates.

### The PROST index
**Pre-processing.** First, we interpolated irregular spots to regular grid, and considered each gene as a grayscale image. The irregular spots are defined as spots that do not conform to either ordered squares or hexagonal arrangement structures. For 10x Visium datasets, we adopted strategy of one-to-one corresponding to interpolate on the gird. The grid value is equal to the UMI count of the spot at the corresponding position. For large-scale ST datasets such as Stereo-seq, we interpolated input spots set linearly on each grid. After pre-processing, each original gene data with $N$ spots was converted into a grayscale image with $N'$ grids (hereinafter, in the PI (PROST Index) processing, denoted as pixels). Due to the severe noise of ST technologies in gene expression measurements[62], the spatial expression patterns of genes were interfered. We utilized the Min-Max Normalization method to scale the gene expression matrix values between 0 and 1, transforming the adjusted matrix into gene grayscale images. Following this, Gaussian filtering was implemented with default setting to mitigate the influence of outliers and to smooth the pervasive noise within the gene grayscale images.

**Binary segmentation for foreground and background.** In the processing of binary segmentation for foreground and background, a threshold value is adaptively and iteratively selected for each gene grayscale image. To align the foreground with significant expression regions and enhance the precision of subsequent foreground labeling, an image morphological closure operation is performed post-binary segmentation. This step aids in mitigating internal noise. The size of

the convolutional kernel used in the image morphological closure can be set within the software function "PROST.cal_PI()" using the "kernel_size" parameter. The parameter kernel_size is set to 5 for 10x Visium and Slide-seq data, 6 for Stereo-seq data, and 8 for SeqFISH data, which settings are inversely correlated with the resolutions of ST datasets. The foreground labeling is executed using the function "skimage.measure.label()" from the skimage python package (version 0.19.2). The parameter "connectivity=2" is specified, while other parameters retain their default settings. This function in an 8-connectivity way allows pixels are connected if they are adjacent either vertically, horizontally, or diagonally. If pixels are adjacent and not part of the background, they are labeled as foregrounds. Moreover, to mitigate the risk of misclassification stemming from an insufficient number of foreground pixels, the "del_rate" parameter in the PI program can be adjusted. If the count of foreground pixels falls below a specified percentage of the total gene image pixels, that particular gene is omitted. By default, the "del_rate" parameter is set to 0.01, implying that a gene is considered only if the proportion of its foreground pixels exceeds 1% of its total pixels.

After the foreground labeling, we marked all pixels in a connected area of foreground with the same value, and obtained $L$ regions $\mathbf{F} = \{\mathbf{F}_1, \mathbf{F}_2, \cdots, \mathbf{F}_L\}$, $\mathbf{F}_l \in \mathbb{R}^{N_l}$, where $N_l$ is the pixel number of region $\mathbf{F}_l$. At this time, the whole gene image is divided into $L+1$ regions with $L$ foreground and one background. The subregion segmentation of gene image is the basis of PROST Index.

**Calculating PROST Index.** Before design the PROST Index, to ensure genes comparable to each other, we applied max-min normalization to all pixels for each gene grayscale image, with $Y_i$ being the ith pixel value, and $\mu$ be the mean value of the normalized gene image. Let $Y_{li}$ be the ith pixel value of the region $\mathbf{F}_l$, and $Y_{Bi}$ be the ith pixel value of the background $\mathbf{B}$, with $N_B$ being the number of pixels in $\mathbf{B}$. Let

$$\mu_l = \frac{1}{N_l} \sum_{i=1}^{N_l} Y_{li} \tag{1}$$

$$\mu_B = \frac{1}{N_B} \sum_{i=1}^{N_B} Y_{Bi} \tag{2}$$

be the mean values of the foreground region $\mathbf{F}_l$ and background $\mathbf{B}$, respectively. Let

$$\sigma_l^2 = \frac{1}{N_l} \sum_{i=1}^{N_l} (Y_{li} - \mu_l)^2 \tag{3}$$

$$\sigma^2 = \frac{1}{N'} \sum_{i=1}^{N'} (Y_i - \mu)^2 \tag{4}$$

be the variances of the foreground region $\mathbf{F}_l$ and the considered gene image, respectively. Recall that $N'$ is the total number of pixels in the gene image. With these definitions in mind, to design the index, we first propose two factors: a *Significance* factor and a *Separability* factor for PI. Specifically, on the one hand, we define the *Significance* factor, assuming that spatially variable genes (SVG) images would exhibit significant differences between regions of foreground and background, and the pixel values in a connected area should have less dispersion, which is then given by

$$Significance = \frac{\frac{1}{L}\sum_{l=1}^{L}(\mu_l - \mu_B)}{\sum_{l=1}^{L}\frac{\sigma_l}{\mu_l}} \tag{5}$$

On the other hand, we define the *Separability* factor, assuming that there are large separability between regions of foreground and background in SVG images, which is then given by

$$Separability = 1 - \frac{\sum_{l=1}^{L} N_l \sigma_l^2}{N' \sigma^2} \tag{6}$$

Notice that the *Significance* factor aims at identifying the region with more homogeneity and less dispersion, indicating genes with significant spatial expression. While, the *Separability* factor is inspired by the concept of spatial stratified heterogeneity[63], where gene expression is homogeneous within each region but not between regions, revealing the existence of distinct mechanisms in different regions.

Finally, we model the PROST Index as follows:

$$PROST\ Index = f(Significance, Separability) \tag{7}$$

Specifically, $f$ is defined as:

$$PROST\ Index = \psi\left[\frac{\frac{1}{L}\sum_{l=1}^{L}(\mu_l - \mu_B)}{\sum_{l=1}^{L}\frac{\sigma_l}{\mu_l}}\right] \times \psi\left(1 - \frac{\sum_{l=1}^{L} N_l \sigma_l^2}{N'\sigma^2}\right) \tag{8}$$

where $\psi(\bullet)$ is normalization operation in gene dimension. As can be observed, higher *Significance* value represents more homogeneity within a region, and higher *Separability* value represents more heterogeneity among regions. Combined by the two factors, PI score can well quantify the spatial patterns for a given gene.

## Hypothesis testing

**Interpretation.** PROST Index (PI) is an inferential statistic, which means that the results of the analysis are always interpreted within the context of a null hypothesis. As a spatial pattern analysis tool specifically developed for ST, PI shares the same null hypothesis with spatial autocorrelation statistics, i.e., complete spatial randomness (CSR)[64]. CSR states that the spatial pattern of gene expression is generated from some random process. Rejecting the null hypothesis indicates that the gene spatial expression exhibits statistically significant clustering or dispersion, ensuring the quantization value of PI exhibits statistically significant rather than being generated from a random spatial process. Thus, we performed a hypothesis testing method for spatial autocorrelation statistics to help determine whether the null hypothesis (CSR) should be rejected.

**Parametric test.** Specifically, take a representative spatial autocorrelation statistic Moran's $I$ as an example. The expected value is $E(I) = -1/(N-1)$ and the variance $V(I)$ is calculated differently under an assumption of randomness[17] versus an assumption of normality[65]. These two assumptions represent the theoretical way to produce gene spatial expression under the hypothesis of randomly placing gene spatial expression. The randomization assumption postulates that the observed spatial pattern of gene expression is one of many possible spatial arrangements. The gene expression values are fixed, and only the spatial location changes. While the normalization assumption states that the gene expression and their locations are one of many possible random samples, while neither the gene expression values, nor the spatial locations are fixed. Next, since Moran's $I$ approximately follows a normal distribution[66], its value may be assessed by the $z$-score of the normal distribution:

$$z = \frac{I - E(I)}{\sqrt{V(I)}} \tag{9}$$

With $z$-score of an observed value, a one-tailed $p$-value can be calculated as:

$$p_\alpha = \Pr(Z \leq z\ or\ Z \geq z) = 1 - \Phi(|z|) \tag{10}$$

where $\Phi(\bullet)$ is the cumulative distribution function (CDF) of standard normal distribution:

$$\Phi(x) = \frac{1}{\sqrt{2\pi}} \int_{-\infty}^{x} e^{-\frac{x^2}{2}} dx \qquad (11)$$

**Nonparametric test.** The above inferences based on the expectation and variance employ an approximation to a standard normal distribution, which may not be valid when the underlying assumptions are not met. Thus, we further provide a computational approach based on permutation test[67]. The method first randomly permutes the observations (gene expression values) over the locations to calculate a reference distribution for the statistic, then a pseudo $p$ value can be calculated as:

$$p_{\beta} = \frac{R+1}{D+1} \qquad (12)$$

where $R$ is the number of times the Moran's $I$ computed from the permuted gene expression is greater than or equal to the initial statistic, and $D$ is the number of permutations, which is typically taken as 999. It is worth noting that the pseudo $p$ value is only a summary of the reference distribution, and the extent of significance is determined by the number of permutations.

The computational cost of the permutation tests primarily stems from the repetitive calculation of Moran's Index. To tackle this challenge, we implemented matrix computation methods and devised a function named *batch_morans_I()* to effectively mitigate the redundancy of these computations. Furthermore, recognizing that the calculations between genes are mutually exclusive, we leveraged the process pool technique within the multiprocessing library to achieve data parallelism, further decreasing the overall computational overhead.

**Multiple testing correction.** As the number of tests increases, the probability of mistakenly rejecting the null hypothesis (i.e., the Type I error rate, or false-positive rate) greatly increases. Therefore, we applied the Benjamini–Hochberg method to control the False Discovery Rate (FDR) for multiple testing correction[68]. A very small FDR implies that the observed spatial patterns are unlikely to arise from a random process and therefore the null hypothesis (CSR) can be rejected.

### The PROST neural network

**Pre-processing.** In the pre-processing stage, the spots outside the tissue area and genes expressed in less than ten spots were removed. Next, normalization was performed. The UMI count for each gene was divided by the total UMI count in a given spot for normalization. Then, the gene expression values were transformed to a natural log scale. Finally, we selected SVGs with top 3000 PI scores as the input.

**Construction of gene expression graph.** After pre-processing, PROST converted the spatial location information into a directed graph $G(V, E)$. This graph was built based on the spatial proximity of spots, where each vertex $vV$ represents a spot. For a vertex $v_i \in V$, it only connects to its $k$ adjacent neighbors. Let $\mathcal{N}_i$ be the set of $v_i$'s $k$ nearest neighbor vertexes. The topology structure of graph $G$ can be denoted by an adjacency matrix $\mathbf{A} = \{a_{ij}\} \in \mathbb{R}^{N \times N}$, where $a_{ij} = 1$ if $v_j \in \mathcal{N}_i$, indicating that there is an edge from vertex $v_i$ to vertex $v_j$, else $a_{ij} = 0$. $\mathbf{D} = \mathrm{diag}(d_1, d_2, \cdots, d_N) \in \mathbb{R}^{N \times N}$ denotes the degree matrix of $\mathbf{A}$, where $d_i = \sum_{v_j \in \mathcal{N}_i} a_{ij}$ is the degree of node $v_i$. With the renormalization trick[69], we took self-looped adjacency matrix $\hat{\mathbf{A}} = \mathbf{I} + \mathbf{A}$ as the gene expression graph, with $\mathbf{I} \in \mathbb{R}^{N \times N}$ being the identity matrix.

**Laplacian smoothing.** First, principal component analysis (PCA) was performed on the gene expression matrix, and the top 50 principal components (PCs) $\mathbf{X}' \in \mathbb{R}^{N \times 50}$ were selected for subsequent calculations. Then, we adopted a low-pass denoising operation to aggregate neighbor information. Due to the fact that ST data is with high sparsity, here we propose to use the graph Laplacian smoothing filter $\mathbf{H}$, which, able to achieve the same effect as graph convolution operation[70], is defined as:

$$\mathbf{H} = \mathbf{I} - \gamma \widetilde{\mathbf{L}} \qquad (13)$$

where $\gamma$ is a real value and set to 2/3 by default; $\widetilde{\mathbf{L}}$ denotes the symmetric normalized graph Laplacian matrix, which is defined as:

$$\widetilde{\mathbf{L}} = \hat{\mathbf{D}}^{-\frac{1}{2}} \hat{\mathbf{L}} \hat{\mathbf{D}}^{-\frac{1}{2}} \qquad (14)$$

where $\hat{\mathbf{D}}$ and $\hat{\mathbf{L}}$ are the degree matrix and the Laplacian matrix corresponding to $\hat{\mathbf{A}}$. Here, $\hat{\mathbf{L}}$ is calculated as:

$$\hat{\mathbf{L}} = \hat{\mathbf{D}} - \hat{\mathbf{A}} \qquad (15)$$

Finally, we stacked up $t$ Laplacian smoothing filters as follows:

$$\widetilde{\mathbf{X}} = \mathbf{H}^t \mathbf{X}' \qquad (16)$$

where $\widetilde{\mathbf{X}} \in \mathbb{R}^{N \times 50}$ denotes the Laplacian smoothed feature matrix, and $t$ was set to 2. Through stacked Laplacian smoothing filter, high-frequency noises in the gene expression were effectively filtered.

**Graph attention network.** To make the model acquire the capability of adaptive learning the importance between neighboring vertices, we applied self-attention mechanism on the graph structure. The key to the attention mechanism is the graph attention layer[71]. The input to this layer is the smoothed feature matrix $\widetilde{\mathbf{X}}$ consisting of all vertex features $\mathbf{h} = \{\mathbf{h}_1, \mathbf{h}_2, \cdots, \mathbf{h}_N\}, \mathbf{h}_i \in \mathbb{R}^{50}$, where $\mathbf{h}_i$ represents the Laplacian smoothed gene expression feature of the ith spot. Recall that $N$ represents the number of spots. Besides, based on the spatial autocorrelation hypothesis that near vertices are more closely related to each other[72], in this paper, the first-order neighbors of vertices were considered when calculating attention coefficients. The attention coefficients of vertices $v_i$ and $v_j$ can be expressed as:

$$e_{ij} = r\left( \left[ \mathbf{W}\mathbf{h}_i || \mathbf{W}\mathbf{h}_j \right] \right) \qquad (17)$$

where $e_{ij}$ indicates the importance of $v_j$'s features to $v_i$; $\mathbf{W}$ denotes a linear transformation of the vertex features with shared parameters, which is a trainable weight matrix; $[\cdot || \cdot]$ denotes a concatenate operation on the transformed features of vertices $v_i$ and $v_j$; $r(\cdot)$ denotes a shared attention mechanism for mapping the concatenated high-dimensional features to a real number. In this paper, $r(\cdot)$ is a single-layer feedforward neural network, parametrized by a weight vector $\mathbf{w}$, using LeakyReLU as the nonlinear activation function. The attention coefficients $e_{ij}$ can be further expressed as:

$$e_{ij} = \mathrm{LeakyReLU}\left( \mathbf{w}^{\mathrm{T}} \left[ \mathbf{W}\mathbf{h}_i || \mathbf{W}\mathbf{h}_j \right] \right) \qquad (18)$$

Then, the softmax function is introduced to normalize the correlation coefficient $e_{ij}$ between $v_i$ and its neighbors $\mathcal{N}_i$:

$$\vartheta_{ij} = \frac{\exp\left( e_{ij} \right)}{\sum_{j=1}^{k} \exp\left( e_{ij} \right)}, v_j \in \mathcal{N}_i \qquad (19)$$

Finally, the linear combination of the corresponding features is calculated using the normalized attention coefficients $\vartheta_{ij}$. The features are weighted and summed by the nonlinear activation function,

resulting the final output feature vector of $v_i$:

$$\mathbf{h}_i' = g\left(\sum_{v_j \in \mathcal{N}_i} \vartheta_{ij} \mathbf{W}\mathbf{h}_i\right), \mathbf{h}_i' \in \mathbb{R}^{50} \qquad (20)$$

where $\mathbf{h}_i'$ is the embedding feature that $v_i$ aggregates information of its neighbors; $g(\cdot)$ is a nonlinear activation function such as elu[73]. By simultaneously updating parameters $\mathbf{W}$ and $\mathbf{w}$, a set of low-dimensional latent representations $\mathbf{h}' = \{\mathbf{h}_1', \mathbf{h}_2', \cdots, \mathbf{h}_N'\}, \mathbf{h}_i' \in \mathbb{R}^{50}$, integrating gene expression and spatial information from neighborhood, can be achieved.

**Unsupervised clustering.** We performed self-supervision mechanism for clustering enhancement[74] on the embedding features. First, we initialized the clustering centers by different approaches. For data with a given number of the clusters, we suggest using the mclust[75] or $k$-means algorithm to obtain clusters with a specific number of clustering centers. Otherwise, we suggest using the louvain[23] or leiden[76] algorithm, manually adjusting resolutions to obtain a series of different clustering centers. After initialization, the corresponding clustering centers are denoted by $\{\varphi_1, \varphi_2, \cdots, \varphi_K\}$, where $K$ is the number of clusters. After initialization, the clusters were updated through a self-learning process via two steps. In the first step, we used the Student's $t$-distribution as the kernel to measure the similarity (distance) $\mathbf{Q}$ between $\mathbf{h}_i'$ and $\varphi_k$:

$$q_{ik} = \frac{\left(1 + ||\mathbf{h}_i' - \varphi_k||^2/\nu\right)^{-\frac{\nu+1}{2}}}{\sum_{k'=1}^{K}\left(1 + ||\mathbf{h}_i' - \varphi_{k'}||^2/\nu\right)^{-\frac{\nu+1}{2}}} \qquad (21)$$

where $\nu$ denotes the degrees of freedom of Student's $t$-distribution, which is chosen to be 0.5 in this study. Then, we define an auxiliary target distribution $\mathbf{P}$ on the basis of $q_{ik}$:

$$p_{ik} = \frac{q_{ik}^2/\sum_{i=1}^{N}q_{ik}}{\sum_{k'=1}^{K}\left(q_{ik'}^2/\sum_{i=1}^{N}q_{ik'}\right)} \qquad (22)$$

which measures the probability of $v_i$ belongs to the kth cluster, and can be used to measure the confidence of spots' clustering assignment. To make the embedding feature closer to the cluster centers, we used Kullback−Leibler (KL) divergence loss as the objective function to iteratively assign spots in higher confidence clusters:

$$L = KL(\mathbf{P}|\mathbf{Q}) = \sum_{i=1}^{N}\sum_{k=1}^{K}p_{ik}\log\frac{p_{ik}}{q_{ik}} \qquad (23)$$

In the second step, we used Adam algorithm[77] to minimize $L$ by jointly updating the clustering centers $\varphi_k$, $\mathbf{W}$ and $\mathbf{w}$. The gradients of $L$ with respect to embedding feature $\mathbf{h}_i'$ and each clustering centers $\varphi_k$ are computed as:

$$\frac{\partial L}{\partial \mathbf{h}_i'} = \frac{\nu+1}{\nu}\sum_{k=1}^{K}\left(1 + \frac{||\mathbf{h}_i' - \varphi_k||^2}{\nu}\right)^{-1} \times (p_{ik} - q_{ik})(\mathbf{h}_i' - \varphi_k) \qquad (24)$$

$$\frac{\partial L}{\partial \varphi_k} = \frac{-(\nu+1)}{\nu}\sum_{i=1}^{K}\left(1 + \frac{||\mathbf{h}_i' - \varphi_{k'}||^2}{\nu}\right)^{-1} \times (p_{ik} - q_{ik})(\mathbf{h}_i' - \varphi_k) \qquad (25)$$

where $\frac{\partial L}{\partial \mathbf{h}_i'}$ are then passed on to the PNN and used in backpropagation to compute $\frac{\partial L}{\partial \varphi_k}$. During the training process, the target distribution $p_{ik}$ can be treated as "ground-truth". It also depends on the current soft

assignment $q_{ik}$ which updates at every iteration. To avoid instability in the training process caused by the constant change of the target that would obstruct convergence, $p_{ik}$ is updated in every three iterations using all embedded nodes, and the label of the ith spot could be obtained by:

$$s_i = \arg\max_k q_{ik} \qquad (26)$$

**Post-processing (optional).** After clustering, we also provided a post-processing tool to optimize the clustering results. Under the condition of retaining local structure, this tool utilizes morphological information to effectively remove the trivial points inside the clusters, so that improving the spatial smoothness. Specifically, the tool first collects cluster label information within a certain neighborhood, then encourages spatially near spots to belong to the same category by reassigning cluster labels.

### Spatial trajectory inference
To show the spatial trajectory compared with UMAP, the *scanpy.pl.paga_compare()* function in the SCANPY package[36] (v1.8.2) was employed, and UMAPs as well as PAGA graphs were visualized respectively.

### Comparison with other methods
**Domain segmentation.** To benchmark domain segmentation performance, we compared PROST with seven state-of-the-art methods, consisting of SCANPY[36], stLearn[24], SpaGCN[25], BayesSpace[27], SpaceFlow[33], STAGATE[32] and BASS[28] on the 10x Visium human dorsolateral prefrontal cortex (DLPFC) ST dataset (12 sections)[37]. To make the domains comparable between benchmarking methods, we set the target number of clusters equal to the number of clusters in the annotation for all methods. Concretely, for sections with id 151669, 151670, 151671, and 151672, the target number of clusters was set to 5, and for the others, it was set to 7. We used Adjusted Rand Index (ARI) and Normalized Mutual Information (NMI) to quantify the similarity between the clustering results and the annotation.

For SCANPY analysis, genes expressed in less than three spots were filtered, then gene expressions were normalized and log-transformed. Totally, 3000 top highly variable genes were selected as the inputs of PCA processing, and the first 30 PCs were used. Next, the nearest neighbor network was constructed using the *scanpy.pp.neighbor()* function with default parameters. At last, the *scanpy.tl.leiden()* function that implemented the Leiden algorithm was applied to obtain clustering assignments, with manual adjustment of resolution parameters to achieve the target number of clusters.

With stLearn, the input was the 10x Visium original data and the corresponding histology image. Then, the function *stlearn.pp.filter_genes()* was executed with *min_cells = 3*, and successively run the following functions *stlearn.pp.normalize_total()*, *stlearn.pp.log1p()*, *stlearn.pp.tiling()*, *stlearn.pp.extract_feature()*. Next, the first 15 PCs were selected using *stlearn.pp.run_pca()*. Then, the *stlearn.spatial.SME.SME_normalize()* function was used with the parameter setting *use_data = "raw"* and *weights = "physical_distance"*. Last, *stlearn.pp.scale()* and *stlearn.em.run_pca()* functions were employed with the first 15 PCs. The low-dimensional embeddings were used for clustering by the $k$-means algorithm.

The HMRF algorithm was implemented by the Giotto[19] package. First, we created a Giotto object using the gene expression matrix and position information, and the low expression genes were filtered out. Gene expressions were normalized with the default scale factor. Next, the top 100 genes with consistent spatial expression patterns were selected using BinSpect-kmeans. Finally, the spatial network was constructed using the *createSpatialNetwork()* function with the Delaunay

method, and the *doHMRF()* function was employed, resulting in the specified number of clusters.

With SpaGCN, the genes expressed in less than three spots were eliminated. The gene expressions were normalized, multiplied by 10,000, and then transformed to a natural log scale. Next, the adjacency matrix was calculated using a histology image, with a setting of $s = 1, b = 49$, and $p = 0.5$. The resolution parameter of Louvain algorithm was automatically searched according to the target number of clusters. Finally, the cluster results were refined.

For BayesSpace analysis, we created an object with spots' row coordinates, column coordinates and gene expression matrix through the *SingleCellExperiment()* function. Then, the *spatialPreprocess()* function was executed with setting of *platform = "visium"* as well as the selection of the 2000 top HVGs (Highly Variable Genes) and first 15 PCs. Finally, we set the target number of clusters and *init.method = "mclust"* for *spatialCluster()* function to obtain a certain target clustering number.

As for SpaceFlow, we followed the settings of the tutorial in its package. Specifically, a SpaceFlow object was created using *SpaceFlow.SpaceFlow()* method after reading in 10x Visium original data. The 3000 top highly variable genes were selected as the inputs. Then, we used *train()* method to train the model, and the generated embeddings was used for clustering with *segmentation()* method. The parameters in the above procedures were the same as those in the tutorial of code availability section in the original paper[33].

With STAGATE analysis, the data preprocessing step was the same as SCANPY, starting with log-normalization and selecting 3000 top highly variable genes. Next, we used *STAGATE.Cal_Spatial_Net()* and *STAGATE.Stats_Spatial_Net()* functions to construct a spatial neighbor network with the setting of *rad_cutoff = 150*. In training model step, the function parameter $\alpha$ was set to 0. Last, the embeddings generated by training model were used for mclust[75] algorithm to cluster.

For BASS, the gene expression matrix and spatial locations were input to create a BASS object. To ensure consistency of algorithms across all datasets, only one section at a time will be entered for analysis. Then, we used the preprocess function, which includes log-normalization, selecting 2000 SVGs by SPARK-X, and extracting the top 20 PCs with PCA. Next, we ran the main program of BASS and post-processed the results using the default parameters.

**Spatially variable genes identification.** To benchmark the performance of SVG identification, we calculated Moran's *I* and Geary's *C* to quantify the SVGs detected by PROST and five methods, consisting of Seurat[40], SpatialDE[12], SPARK-X[41], SINFONIA[19] and scGCO[14].

With Seurat, we took the original ST data including spatial information and gene expression as input to create a Seurat-Object. The gene expression was log-normalized and multiplied by 10,000. Then, we used *FindVariableFeatures()* function with the method of *"vst"*. Finally, the top 3000 highly variable genes were selected.

With SpatialDE, the input included spatial information and gene expression. The genes that expressed in less than three spots were filtered. Then we performed *NaiveDE.stabilize()* function to convert the data to normal distribution. Next, a linear regression relationship between sample size and gene expression was established by *NaiveDE.regress_out()* function. At last, *SpatialDE.run()* function was employed to generate *p*-values for spatial differential expression.

For SPARK-X analysis, we input gene expression and location information to the *sparkx()* function with the set of *option = "mixture"*, with outputs of combined *p*-values and adjusted *p*-values.

As for SINFONIA, genes expressed in less than three spots were eliminated, and then gene expressions were log-normalized. Next, we employed the *sinfonia.spatially_variable_genes()* function with default parameters to identify SVGs.

For scGCO, we used the *read_spatial_expression()* to read spatial information and filter the gene expression, then the data was normalized by *normalize_count_cellranger()*. Subsequently the FDR of each gene was calculated by *create_graph_with_weight()*, *multiGMM()*, *identify_spatial_genes()*. It should be noted that default parameters were maintained for all aforementioned functions.

**Statistics and reproducibility**
In this study, no statistical method was used to predetermine sample size. All data used in this study were collected from public resources and used to demonstrate the functionalities and performance of PROST. We performed quality control of spatial resolved transcriptomics data based on the commonly used and pre-established criteria in this field. Thus, no data were excluded from the analyses. The experiments were not randomized. Analyses were conducted exclusively on published data, as documented in their original publications, precluding blinding by investigators during reanalysis.

**Reporting summary**
Further information on research design is available in the Nature Portfolio Reporting Summary linked to this article.

## Data availability
All data analyzed in this study were downloaded in raw form from the original publications. (1) The 10x Visium DLPFC[37] data is available at LIBD (http://research.libd.org/spatialLIBD); (2) The other datasets generated by 10x Visium platform are available at the 10x Genomics website (https://support.10xgenomics.com/spatial-gene-expression/datasets); (3) The ST sequencing data used in this study is available in the SpatialDB database:[78] (http://www.spatialomics.org/SpatialDB); (4) The processed Stereo-seq data from mouse olfactory bulb tissue is accessible at SEDR[29] (https://github.com/JinmiaoChenLab/SEDR_analyses); (5) The mouse somatosensory cortex dataset[79] is available from (http://linnarssonlab.org/osmFISH/availability/); (6) The Slide-seq data[7] is download from (https://singlecell.broadinstitute.org/single_cell/study/SCP354/slide-seq-study); (7) The Slide-seqV2 datasets[80] are available at the Broad Institute Single Cell Portal (https://singlecell.broadinstitute.org/single_cell/study/SCP815/highly-sensitive-spatial-transcriptomics-at-near-cellular-resolution-with-slide-seqv2); (8) Simulated gene expression counts[81] are available at (https://github.com/acheng416/Benchmark-CTCM-ST); (9) The processed SeqFISH mouse embryogenesis dataset[50] with segmentation information and associate metadata are available at https://crukci.shinyapps.io/SpatialMouseAtlas/. Source data are provided with this paper.

## Code availability
The PROST algorithm is implemented in Python and is available on GitHub (https://github.com/Tang-Lab-super/PROST) and Zenodo[82].

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

## Acknowledgements

This work was supported by grants from the National Natural Science Fund of China (no. T2225019 to J.L.), the National Key R&D Program of China (no. 2020YFA0803300 to Z.T.) and Key-Area Research and Development Program of Guangdong Province (no. 2023B1111020007 to Z.T.)

## Author contributions

Y.L. developed and implemented the PROST algorithm. J.L. and Z.T. conceived, designed, and supervised the project and contributed equally to this work. G.S., R.C., Y.Y., Z.X., L.Y., Y.H., Q.S., and L.W. performed simulations and analyzed real data. Y.L., J.L., and Z.T. wrote the manuscript, with all authors contributing to the revision of the manuscript.

## Competing interests

The authors declare no competing interests.
