## [Peer Review File · Nature Communications]

PROST: quantitative identification of spatially variable genes and domain detection in spatial transcriptomicsReviewer #1 (Remarks to the Author):

Major comments:

1. The author designed a PROST index to capture features from segmented spatial expression data, followed by a neural network to identify spatial domains. Although the idea is interesting, it somehow defeats the purpose of deep neural networks, a key advantage of which is to perform automatic feature extraction/learning. This is especially questionable given the simplicity of the PROST index, where information lost could be inevitable. To justify the approach, the author should analyse datasets using only the NN part of the PROST pipeline and demonstrates that the addition of PROST index is advantageous over deep NN operating directly on original data.
2. I found the lack of statistical justification is rather alarming. Given the noisy nature of the single-cell sequencing data and the test being performed at genome-scale, proper control of false discovery rate is essential. For example, simply using an index score $>$ threshold is rather arbitrary. The PROST index is like classical clustering strategies such as Fisher's discriminant analysis, and a statistically significance procedure should be designed for PROST index to better identify SV genes. Moreover, the domain identification results of PROST should also be measured statistically, which will help to control false discovery rate and make the results more interpretable.
3. Parameters of the method should be further optimised or described in a greater detail. For example, segmentation of foreground and background will have a large impact on the performance, yet only a simple threshold method is used. There is a large body of literatures on image segmentation, and the authors should at least try to optimise this step. The authors generate graphs using k neighbours, but the value of k and how to determine k was never described.
4. MRF represents a prominent strategy to identify tissues domains, for examples works by Zhu et al, Nat. Biotechnol. 36, 1183–1190 (2018), which is completely omitted. These methods should be included and compared with proposed method.
5. A key advantage claimed by the authors is domain segmentation. However, only one dataset is utilized to evaluate domain segmentation performance, given the complexity of biological tissues, especially in disease conditions, this is clearly not sufficient.
6. Standard metrics to measure algorithm performance such as sensitivity, specificity etc are not provided. These should be calculated either on real biological dataset (such as data with known domain annotations) or on simulated datasets. Simulated datasets are also essential to evaluate algorithms' performance under noisy conditions, which is intrinsic for single-cell data.
7. As single-cell data grows into millions of cells per dataset, scalability is essential for analysis strategies. The author should compare scalability of their methods against state-of-the-art methods. This is important because they utilise Laplacian smoothing and operations on large matrix might cause scalability issues.

Minor comments:

1. The authors need to be accurate on statements. The SOMED method is based on SOM and Gaussian process not on deep learning.
2. Irregular spots are not defined.
3. The writing needs to be improved and grammar errors should be corrected.

Reviewer #2 (Remarks to the Author):

Liang et al proposed a method PROST to address two questions in spatial transcriptomics data analysis: the detection of spatial variable genes (SVGs) and the spatial domain clustering or segmentation. They defined a PROST Index as the measurement of the degree of spatial variation of genes, and developed a neural network approach (PNN) based on both the spatial and transcriptional information for clustering. While the idea has some novelty, there are several concerns need to be addressed before the advantage of the proposed method can be agreed upon.

1. The definition of PROST index (PI), although has some intuitive explanation, is quite arbitrary and need further justification. It's stability when being applied on different data also need to be verified. A basic step in PROST is the separation of foreground and background regions based on the intensity of signals. There are several key factors that can easily change the result of this step,

- but the authors didn't give any discussion on what they did in fixing those factors and the reason behind. Also, the authors used PI score >0 as the criterion for defining SVGs. What is the reason for this choice of threshold? How this choice may affect the downstream analysis?
2. This actually rooted in the fundamental question about how to define meaningful spatial variations. The authors seem to abandoned the idea of statistical test as SVGs may not be a yes-or-no thing, but rather a matter of degree. But they should justify that their PI score is a good measure of the degree, and discuss the underlying assumption on SVGs behind this measurement.
 3. The comparison with more advanced clustering methods for spatial clustering is needed. The author said PROST gained the best median ARI score of 0.481 on the DLPFC dataset, but the BASS method (<https://doi.org/10.1186/s13059-022-02734-7>) achieved a similar performance in single section mode and achieved a 0.51 median ARI score in the integrative mode. Could the author compare this method and shows the advantages of PROST?
 4. PNN used graph neural network and unsupervised clustering strategy, just like the previous SpaGCN work, but PROST got a much higher performance as shown in Fig.2 a. I noticed that PROST used the SVG identified by PI score while other methods did not. Does it the main contribution to higher performance? More experiments on the PNN without SVG selection are needed.
 5. The results on the DLFPC dataset showed that STAGATE and SpaceFlow were not the second and third-place methods. Why did the author compare the PROST with their results on the next dataset?
 6. The author said on the mouse embryo data, the spatial domains for the forebrain, midbrain, and hindbrain were intermingled together in the original annotation. However, in the original paper (<https://doi.org/10.1038/s41587-021-01006-2>), they showed the brain subcluster results in Figure 2. The authors should compare the PORST results with the original paper results.
 7. For the "overview of PROST" section, more details about method design ideas need to be explained. For instance, the authors should give more illustrations of two basic metrics in the PI score.
 8. Some wrong legend references exist in the manuscript. For instance, "matched with the manual annotation (Fig.1b)".

Reviewer #3 (Remarks to the Author):

In this manuscript, Liang et al. proposed a quantitative pattern recognition framework named PROST for identifying spatially variable genes and detecting spatial domains in spatial transcriptomics. PROST includes an indicator (PI) without any statistical hypothesis for evaluating variations in spatial patterns of gene expressions and a neighborhood-based graph (PNN) with a self-attention mechanism to integrate spatial and transcriptional information. The authors compared the performance of PROST with several methods on ST datasets generated by different platforms and showed that PROST provides superior performance in both SVG identification and domain segmentation. Overall, the manuscript is well written and organized. While I enjoy reading the paper and appreciate its many potential merits, it would be better if the following concerns could be addressed.

1. With the development of advanced high-throughput sequencing technologies, both efficiency and scalability are essential for computational methods. It would be better if the authors could further benchmark the running time and peak memory usage of different methods and discuss what kind of computing conditions PROST requires.
2. The authors have demonstrated the superior performance of the combination of PI and PNN. It would be interesting if the authors can conduct more model ablation experiments. For example, using the same number of SVGs, what if PI is replaced with state-of-the-art SVG identification methods, such as SINFONIA (DOI: 10.3390/cells12040604)? What if PNN is replaced with state-of-the-art spatial domain identification methods, such as STAGATE and Squidpy (DOI: 10.1038/s41592-021-01358-2)?
3. It is good to quantify the improvement, if any, e.g. using a statistical test to show the improvement is significant, even PROST provided a higher median ARI in Fig. 2a.

4. The limitations of low capture efficiency and high dropouts in scRNA-seq are inherited by most ST technologies. It would be better if the authors could further investigate the robustness of PROST to noise, such as randomly dropping out the entries in the expression matrix to zeros.
5. The authors have provided technical details for the implementation of baseline methods. It would be better if the authors could summarize the parameters used for benchmarking, since some results may be confusing (e.g. STAGATE, DOI: 10.1038/s41467-022-29439-6). For example, if the pipeline and parameters used are the default setting of the original method? What is the rationale if not?
6. The authors claimed that PROST also recognized clear boundaries between layers (Fig. 2a), although its performance (ARI = 0.59) is lower than BayesSpace, exhibiting the highest ARI score of 0.72. I guess it is because ARI may not be the best metric for the evaluation here. The authors could discuss the applicable scenarios and advantages of ARI and NMI, referring to DOIs 10.1038/s41467-021-22495-4 and 10.1093/bioinformatics/btab298.
7. The PAGA graph provided by PROST in Supplementary Figure 1b is interesting. There since to be connections between different layers. Could this possibly reveal some biological insights? I am happy to see more discussion on this.
8. To make the domains comparable between benchmarking methods, the authors set the target number of clusters equal to the number of clusters in the annotation for all methods. However, in most of the real scenarios, we do not know the exact number of spatial domains. Therefore, it would be better if the authors could further benchmark the performance using Louvain or Leiden with default resolutions (DOI: 10.1186/s13059-017-1382-0).
9. It is good to evaluate the clustering performance on the seqFISH data given that the dataset was annotated with tissue structures.
10. A great number of scRNA-seq and ST data have been accumulated in various repositories. Incorporating reference data in analyzing single-cell data can better tackle the high level of noise and technical variation and has been successfully applied in the analysis of various single-cell genomic data (DOIs 10.1038/s41467-021-22495-4, 10.1038/ng.3818, 10.1093/bioinformatics/btab298, 10.1101/2022.11.09.515447). It would be better if the authors could discuss the future direction to incorporate reference data based on PROST in the Discussion section.
11. The authors claimed that PI enables prioritizing spatial variations in gene expression patterns, facilitating explorations of biological insights. It would be better if there is an elaborated tutorial for this.
12. The authors provide a detailed introduction to implementing PROST and extensive Notebook demos. It will be more helpful and convenient if they can provide a Readthedoc page like Squidpy (DOI: 10.1038/s41592-021-01358-2). Besides, they can provide the source codes for reproducing the results in the manuscript.
13. Typos: on lines 20, 23, and 25 of page 4, "Fig.1b", "Fig.1c", and "Fig.1b" should be "Fig.2b", "Fig.2c", and "Fig.2b". Same for "Fig.1d" in the last paragraph of page 4.

Reviewer #1:

Major comments:

1. The author designed a PROST index to capture features from segmented spatial expression data, followed by a neural network to identify spatial domains. Although the idea is interesting, it somehow defeats the purpose of deep neural networks, a key advantage of which is to perform automatic feature extraction/learning. This is especially questionable given the simplicity of the PROST index, where information lost could be inevitable. To justify the approach, the author should analyse datasets using only the NN part of the PROST pipeline and demonstrates that the addition of PROST index is advantageous over deep NN operating directly on original data.

Response:

We appreciate the reviewer's insightful suggestions and understand the concerns raised about the efficiency of domain segmentation when combining our PROST index (PI) with the deep neural network, PNN. The reviewer's comment seems to question the additional value of the PI, considering that deep learning models are intrinsically capable of automated feature extraction and learning.

In response to this concern, we carried out further analysis following the reviewer's advice. We evaluated the domain segmentation results generated by PNN using two different inputs: the complete gene set, and variable numbers of spatially variable genes (SVGs) obtained through distinct methods (**Fig. R1-1**). Our analysis revealed that the incorporation of SVGs selected based on the PI notably improves the accuracy of domain segmentation, as opposed to the case when the PNN is given the entire gene set as input. Our findings further suggest that the accuracy of domain segmentation via PNN increases with the number of SVGs based on the PI, with the most optimal result obtained with 3,000 SVGs. We speculate that the optimal number of SVGs might differ across ST datasets, contingent on the inherent noise of the respective dataset. In addition, we compared the domain segmentation results generated by PNN using SVGs selected by various methods. The outcome of this comparative analysis underscores that the PI outperforms other methods such as Seurat, SINFONIA, and SPARK-X.

In light of these findings, despite its simplicity in calculation, the PI proves to be effective in SVG selection, ultimately yielding better accuracy in terms of domain segmentation. These findings are clarified in a new result section subtitled 'PI-based SVG selection enhances domain segmentation performance of PROST' (lines 293 to 314) and detailed in Supplementary Figure 15 of the revised manuscript.

Fig. R1-1. Performance comparison of PNN combined with various feature selection methods using the 10x Visium DLPFC dataset. The x-axis denotes the different feature selection methods; "None" represents no feature selection, while the numbers following underscores indicate the quantity of features chosen by each respective method.

2. I found the lack of statistical justification is rather alarming. Given the noisy nature of the single-cell sequencing data and the test being performed at genome-scale, proper control of false discovery rate is essential. For example, simply using an index score > threshold is rather arbitrary. The PROST index is like classical clustering strategies such as Fisher's discriminant analysis, and a statistically significance procedure should be designed for PROST index to better identify SV genes. Moreover, the domain identification results of PROST should also be measured statistically, which will help to control false discovery rate and make the results more interpretable.

Response:

We appreciate the reviewer’s constructive suggestions. We agree that establishing an appropriate control for the false discovery rate in PI-based SVG identification is indeed of utmost importance.

In response to this, we developed both parametric and non-parametric tests to evaluate the capacity of PI to identify SVGs. It’s worth noting that PI, functioning as an inferential statistic, aligns with the null hypothesis of spatial autocorrelation statistics, including complete spatial randomness¹. This hypothesis implies that the spatial gene expression pattern is the result of a random process. The rejection of this hypothesis suggests that the spatial expression of the gene displays statistically significant clustering or dispersion, ensuring that the quantization value of PI is statistically significant and not generated from a random spatial process. Thus, we adopted a hypothesis testing method for spatial autocorrelation statistics to evaluate whether the null hypothesis should be rejected. The above parametric test (grounded in expectation and variance) employs an approximation to a standard normal distribution, which may not be valid when the underlying assumptions are not fulfilled. Regarding this concern, we supplemented the parametric approach with non-parametric test based on a permutation test². This method first randomly permutes the observations (gene expression values) over locations to calculate a reference distribution for the statistic, which allows the calculation of a pseudo *p*-value. We then applied the Benjamini–Hochberg method to control the False Discovery Rate (FDR) during multiple testing corrections. For a more detailed description of the implementation of these statistical tests, please refer to the ‘Hypothesis testing’ section (lines 570 to 611) of the method part of the revised manuscript.

We assessed the effectiveness of the implemented statistical tests in SVG identification. Our analysis reveals that the vast majority of SVGs with a PI greater than 0 present statistical significances under either parametric or non-parametric testing, suggesting that PI exhibits both efficiency and robustness in the task of identifying SVGs (**Fig. R1-2**). These findings are clarified at lines 225 to 227 and detailed in Supplementary Figure 13 of the revised manuscript.

Fig. R1-2. Venn diagram shows the intersections between SVGs with PI scores greater than 0 and those with FDR values under 0.05 for the parametric test (a), non-parametric test (b), and both the parametric test and non-parametric test (c). The data from the 10x Visium DLPFC section 151672 was used for the analysis.

3. Parameters of the method should be further optimised or described in a greater detail. For

example, segmentation of foreground and background will have a large impact on the performance, yet only a simple threshold method is used. There is a large body of literatures on image segmentation, and the authors should at least try to optimise this step. The authors generate graphs using k neighbours, but the value of k and how to determine k was never described.

Response:

We appreciate the reviewer's meticulous attention to the details of parameter settings, particularly regarding image segmentation and graph generation within our PROST framework. We fully concur that the choice of parameters can significantly influence the performance of our pipeline, hence the careful consideration of parameter optimization was integral during the development of PROST.

In relation to image segmentation, we have implemented an adaptive thresholding strategy, drawing on classical techniques such as Otsu's method. This strategy considers the spatial distribution of pixel intensities for more refined segmentation, providing superior accuracy in delineating foreground and background elements. Our pipeline then leverages this image segmentation to calculate the PIs. As elucidated in our response to the second question raised by the reviewer, we have shown that PI calculation (based on our image segmentation method) exhibits effectiveness and robustness in SVG identification. This serves to validate the accuracy of our adaptive thresholding strategy for image segmentation. To ensure clarity, we have expanded the description of this strategy in the 'Methods' section of our revised manuscript (lines 513 to 535).

We also extensively tested the key parameters: k -neighbor, minimum distance and max epoch in the process of graph generation. Our analysis reveals that our graph construction procedure exhibits a high degree of robustness to those parameters when dealing with either low-resolution (10x Visium) or high-resolution (osmFISH) spatial transcriptomic data. As shown in **Fig. R1-3a**, the ARI consistently fluctuates between approximately 0.50 to 0.60 when the k -neighbor setting is adjusted from 4 to 12 and the Max epoch parameter ranges from 200 to 800 for the low-resolution 10x Visium dataset. Similarly, as illustrated in **Fig. R1-3b**, the ARI varies within a tight range of approximately 0.60 to 0.68 when the minimum distance setting is adjusted from 600 to 1,000 and the Max epoch parameter ranges from 100 to 1,000 for the high-resolution osmFISH dataset. These findings are now illustrated in Supplementary Figure 32, and the specifics of the parameter settings are detailed in lines 453 to 456 of the revised manuscript, ensuring a comprehensive understanding of the robustness of parameter setting.

Fig. R1-3. 3D contour visualization illustrating the sensitivity of PNN to the parameters: k neighbours, minimum distance, and max epoch parameters, evaluated on the 10x Visium DLPFC (section 151672) dataset (a) and the osmFISH mouse somatosensory cortex dataset (b).

4. MRF represents a prominent strategy to identify tissues domains, for examples works by Zhu et al, Nat. Biotechnol. 36, 1183–1190 (2018), which is completely omitted. These methods should be included and compared with proposed method.

Response:

We appreciate the reviewer for the suggested methods for additional comparative analysis. As recommended, we have incorporated the HMRF method, as detailed in Zhu *et al.*, Nat. Biotechnol., and another method, BASS, for a more comprehensive evaluation of domain segmentation. The

results of this expanded comparative analysis are presented in Figure 2 and Supplementary Figures of the revised manuscript. We believe that the inclusion of these methods enriches our comparative analysis.

5. A key advantage claimed by the authors is domain segmentation. However, only one dataset is utilized to evaluate domain segmentation performance, given the complexity of biological tissues, especially in disease conditions, this is clearly not sufficient.

Response:

We thank the reviewer's comments and agree that a thorough evaluation of domain segmentation performance should be based on a diverse range of datasets. Such datasets should reflect the complexity and variety inherent in biological tissues.

In our original manuscript, we have indeed incorporated multiple datasets derived from various biological tissues. These tissues vary significantly in terms of histological complexity, including human dorsolateral prefrontal cortex, human breast cancer, human lymph node, mouse brain sagittal anterior, mouse olfactory bulb, and mouse kidney, and are represented in spatial transcriptomic data with different resolutions. PROST's performance in domain segmentation has been evaluated across these diverse datasets, with related results illustrated in Supplementary Figures 1-11 and 17-26.

In our revised manuscript, we have expanded our analysis to further assess PROST's domain segmentation performance on mouse somatosensory cortex and cerebellum tissues. These tissues were profiled using osmFISH and Slide-seq techniques, respectively, to generate the spatial transcriptomic data. The outcomes of these additional evaluations are presented in Supplementary Figures 16 and 18.

We underscore PROST's superior performance in domain segmentation when processing a variety of complex tissue-derived datasets at lines 212 to 215 of the revised manuscript. We believe that this expanded analysis substantiates the robustness of PROST in domain segmentation across a broad range of biological complexities.

6. Standard metrics to measure algorithm performance such as sensitivity, specificity etc are not provided. These should be calculated either on real biological dataset (such as data with known domain annotations) or on simulated datasets. Simulated datasets are also essential to evaluate algorithms' performance under noisy conditions, which is intrinsic for single-cell data.

Response:

We appreciate the reviewer's suggestions and agree that simulated datasets are indeed essential for evaluating the performance of our PROST framework under noisy conditions, a common trait of spatial transcriptomic data.

In response to this, we adopted a simulation method that utilizes real spatial locations and simulated gene expression data for generation of semi-synthetic datasets. We applied this method to mouse somatosensory cortex osmFISH and mouse cerebellum Slide-seq datasets, using scDesign³

Fig. R1-4. Assessment of PNN and benchmark methods' performance under varying noise conditions. **a**, Line plot illustrating the performance trends of different methods on the mouse somatosensory cortex osmFISH dataset as data drop-off levels increase. **b**, On the far left, domain annotations of the simulated dataset, juxtaposed with spatial domains identified by diverse methods using this dataset.

based on the procedure described by Cheng *et. al.*⁴. Our analysis reveals that PROST's performance exhibits superior robustness to the sequencing depth, outperforming the benchmark methods (**Fig. R1-4**). These findings are now illustrated in Supplementary Figures 16, and detailed at lines 315 to 321 of the revised manuscript.

7. As single-cell data grows into millions of cells per dataset, scalability is essential for analysis strategies. The author should compare scalability of their methods against state-of-the-art methods. This is important because they utilize Laplacian smoothing and operations on large matrix might cause scalability issues.

Response:

We appreciate the reviewer's perceptive comments and suggestions. We acknowledge the critical requirement for scalability in single-cell/spatial transcriptomic data analysis, particularly as datasets grow to encompass millions of cells.

We indeed recognize that the computation complexity in our method primarily stems from the matrix multiplication operation when calculating the Laplacian matrix from the adjacency matrix during the Laplacian smoothing operation. To mitigate this, we have optimized the computational complexity of the Laplacian smoothing process by introducing sparsity to the matrices involved. Following this optimization, we evaluated the computational performance of PROST in comparison with other benchmark methods (**Fig. R1-5 & R1-6**). Our analysis shows that PROST displays superior performance in terms of both memory usage and CPU time, indicating promising scalability potential for handling large-scale datasets in the future. This result is now clarified at lines 453 to 456 and illustrated in Supplementary Figures 33 and 34 of the revised manuscript.

Fig. R1-5. Evaluation of computational efficiency across various methods using the mouse cerebellum Slide-seq dataset, focusing on runtime and memory consumption. **a**, An efficiency comparison for SVG detection between PI and other methods, including Seurat, SpatialDE, SPARK-X, and SINFONIA. **b**, A juxtaposition of domain segmentation efficiency among PNN, SpaGCN, BASS, SpaceFlow, and STAGATE.

Fig. R1-6. Evaluation of computational efficiency across various methods using the mouse cerebellum Slide-seq dataset, focusing on runtime and CPU usage. **a**, An efficiency comparison for SVG detection between PI and other methods, including Seurat, SpatialDE, SPARK-X, and SINFONIA. **b**, A juxtaposition of domain segmentation efficiency among PNN, SpaGCN, BASS, SpaceFlow, and STAGATE.

Minor comments:

1. The authors need to be accurate on statements. The SOMED method is based on SOM and Gaussian process not on deep learning.

Response:

We appreciate the reviewer's precision and call for accuracy in our descriptions. We have made the necessary correction in the description of these methods in our revised manuscript. These changes can be found in the introduction section at line 58.

2. Irregular spots are not defined.

Response:

We appreciate the reviewer's mention that we omitted the definition of "irregular spots".

These are defined as spots that do not conform to either ordered squares or hexagonal arrangement structures. This clarification has been added to the revised manuscript, specifically at lines 501 to 502.

3. The writing needs to be improved and grammar errors should be corrected.

Response:

We appreciate the reviewer's feedback regarding the need to improve the manuscript's writing and correct grammar errors. In response to this, we have undertaken a thorough review and revision of our manuscript. This includes checking for any grammatical errors, improving sentence construction, and ensuring the clarity and coherence of the narrative.

Last but not least, we gratefully thank the Reviewer again for his/her outstanding comments and suggestions, that greatly helped us to improve the technical quality and presentation of our manuscript.

References:

1. Diggle, P. J. *Statistical Analysis of Spatial Point Patterns*. (Academic Press, 1983).
2. Fisher, R. A. *The Design Of Experiments*. (Oliver and Boyd, 1935).
3. Sun, T., Song, D., Li, W. V. & Li, J. J. scDesign2: a transparent simulator that generates high-fidelity single-cell gene expression count data with gene correlations captured. *Genome Biol.* **22**, 163 (2021).
4. Cheng, A., Hu, G. & Li, W. V. Benchmarking cell-type clustering methods for spatially resolved transcriptomics data. *Brief. Bioinform.* **24**, bbac475 (2023).

Reviewer #2:

Liang et al proposed a method PROST to address two questions in spatial transcriptomics data analysis: the detection of spatial variable genes (SVGs) and the spatial domain clustering or segmentation. They defined a PROST Index as the measurement of the degree of spatial variation of genes, and developed a neural network approach (PNN) based on both the spatial and transcriptional information for clustering. While the idea has some novelty, there are several concerns need to be addressed before the advantage of the proposed method can be agreed upon.

Response:

We appreciate the reviewer's recognition of the novelty of our approach, PROST, in addressing the key questions in spatial transcriptomic data analysis: the detection of SVGs and spatial domain segmentation. We acknowledge the concerns raised by the reviewer and appreciate the opportunity to address them to better illustrate the advantages of our method.

1. The definition of PROST index (PI), although has some intuitive explanation, is quite arbitrary and need further justification. It's stability when being applied on different data also need to be verified. A basic step in PROST is the separation of foreground and background regions based on the intensity of signals. There are several key factors that can easily change the result of this step, but the authors didn't give any discussion on what they did in fixing those factors and the reason behind. Also, the authors used PI score >0 as the criterion for defining SVGs. What is the reason for this choice of threshold? How this choice may affect the downstream analysis?

Response:

We appreciate the reviewer's insightful comment. The reviewer has raised three distinct questions in this comment. We address these questions one by one below.

(1) The reviewer asked about the separation of foreground and background regions based on the intensity of signals and how different factors can influence this step. In relation to image segmentation, we have implemented an adaptive thresholding strategy, drawing on classical techniques such as Otsu's method. This strategy considers the spatial distribution of pixel intensities for more refined segmentation, providing superior accuracy in delineating foreground and background elements. Additionally, we have employed post-processing steps such as morphological opening and closing, following the image segmentation step. These are applied with default settings and effectively reduce noise in the foreground and manage the quantity of foreground elements, thus ensuring the robustness of our image segmentation. Our pipeline then leverages this image segmentation process to calculate the PIs. Our subsequent PI evaluation using different statistical tests, which we detail further in our responses below, affirms the accuracy of our adaptive thresholding strategy for image segmentation. For clarity, we have provided a more detailed description of this strategy in the 'Methods' section of our revised manuscript (lines 513 to 535).

(2) The reviewer asks why a PI score > 0 was used as the criterion for defining SVGs and how this choice might affect downstream analysis. In response to this, we have moved away from using the PI score of > 0 as an arbitrary cutoff and have instead developed statistical tests for PIs in SVG identification. We have introduced both parametric and non-parametric tests for PIs. For the parametric test, PI, acting as an inferential statistic, aligns with the null hypothesis of spatial autocorrelation statistics, specifically complete spatial randomness². Rejecting this null hypothesis indicates statistically significant clustering or dispersion in spatial gene expression. Recognizing that this parametric test may not be reliable if its underlying assumptions are not satisfied, we thus supplement it with a non-parametric permutation test¹. This method first randomly permutes the observations (gene expression values) over the locations to calculate a reference distribution for the statistic, which allows the calculation of a pseudo p -value. For a more detailed description of the implementation of these statistical tests, please refer to the 'Hypothesis testing' section (lines 554 to 595) of the Methods part of the revised manuscript.

We then assessed the effectiveness of the implemented statistical tests in SVG identification. Our analysis reveals that the vast majority of SVGs with a PI greater than 0 present statistical significances under either parametric or non-parametric testing, suggesting that the PI score is both efficient and robust in terms of SVG identification (**Fig. R2-1**).

Fig. R2-1. Venn diagram shows the intersections between SVGs with PI scores greater than 0 and those with FDR values under 0.05 for the parametric test (a), non-parametric test (b), and both the parametric test and non-parametric test (c). The data from the 10x Visium DLPFC section 151672 was used for the analysis.

(3) The reviewer asks for verification of PI's stability when applied to different data sets. In our original manuscript, we have indeed incorporated multiple datasets derived from various biological tissues to evaluate PI's stability in domain segmentation. The related results are illustrated in Supplementary Figures and detailed in the revised manuscript (line 212 to 215). Furthermore, in our revised manuscript, we have expanded our analysis to further assess PI's stability for domain segmentation on osmFISH and Slide-seq datasets. The outcome of these additional evaluations is presented in Supplementary Figures 16 and 26.

Collectively, these comprehensive evaluations provide robust evidence for the stability of PI in domain segmentation when using PROST.

2. This actually rooted in the fundamental question about how to define meaningful spatial variations. The authors seem to abandoned the idea of statistical test as SVGs may not be a yes-or-no thing, but rather a matter of degree. But they should justify that their PI score is a good measure of the degree, and discuss the underlying assumption on SVGs behind this measurement.

Response:

We appreciate the reviewer's astute comments and agree that defining meaningful spatial variation is indeed a foundational question in spatial transcriptomic data analysis. Our aim in developing the PI was not to discard the idea of a statistic test for SVGs, but to provide a quantitative measure that captures the degree of spatial variability.

Underlying the PI score is the assumption that SVGs display significant spatial pattern deviation from complete spatial randomness. To encapsulate this, we introduced two components in the calculation of PIs: the Significance factor and the Separability factor. The Significance factor is premised on the assumption that, in SVG images, there would be notable differences between the foreground and background regions, and that pixel values within a connected area should exhibit less dispersion. This factor is designed to identify regions with more homogeneity and less dispersion, indicative of genes with significant spatial expression. The Separability factor, on the other hand, is

inspired by the concept of spatial stratified heterogeneity, where gene expression is homogeneous within each region but differs between regions, suggesting the presence of distinct mechanisms across different regions. Taken together, these factors enable the PI score to effectively quantify the spatial patterns for a given gene.

In response to the reviewer's comments, we have elaborated on the underlying assumptions of the PI in the revised manuscript (lines 129 to 141). This clarification helps underscore the PI score as a comprehensive measure of the degree of spatial variability.

3. The comparison with more advanced clustering methods for spatial clustering is needed. The author said PROST gained the best median ARI score of 0.481 on the DLPFC dataset, but the BASS method (<https://doi.org/10.1186/s13059-022-02734-7>) achieved a similar performance in single section mode and achieved a 0.51 median ARI score in the integrative mode. Could the author compare this method and shows the advantages of PROST?

Response:

We appreciate the reviewer's astute observation. In our analysis, we indeed observed that BASS achieves a marginally higher median ARI score than PROST (Supplementary Table 3). However, it is important to note that PROST presents less variable ARI scores across twelve DLPFC dataset compared to BASS and other benchmark methods (Fig. R2-2), highlighting the robustness of PROST in domain segmentation performance. Upon a more detailed comparison, PROST emerged with the highest average ARI score (0.474), showcasing superior performance compared to BASS (average ARI=0.456) and other benchmark methods (Supplementary Table 3). Moreover, our analysis demonstrates a slight superiority of PROST over BASS in terms of domain segmentation when dealing with datasets derived from complex tissues at high-resolutions. We illustrate these comparison results in Supplementary Figures 16, 26 and 27 of our revised manuscript.

Fig. R2-2. Boxplot presenting the ARI values, summarizing the domain segmentation accuracy for each method across all 12 sections of the DLPFC dataset.

4. PNN used graph neural network and unsupervised clustering strategy, just like the previous SpaGCN work, but PROST got a much higher performance as shown in Fig.2 a. I noticed that PROST used the SVG identified by PI score while other methods did not. Does it the main contribution to higher performance? More experiments on the PNN without SVG selection are needed.

Response:

We appreciate the reviewer's insightful observations and concerns regarding the integration of our PI with the PNN, and the subsequent impact on domain segmentation performance.

In response to this, we performed additional analyses as suggested by the reviewer. We tested the domain segmentation results generated by PNN using two distinct inputs: the entire gene set and variable numbers of SVGs selected through different methods (Fig. R2-3). Our analyses showed that the accuracy of domain segmentation achieved by PNN significantly improved when utilizing SVGs selected based on the PI, compared to the case when PNN was supplied the entire gene set as input. Moreover, the accuracy of domain segmentation enhanced with the increase in the number of SVGs selected using PI, achieving optimal results with 3,000 SVGs. We speculate that the optimal number of SVGs might differ across ST datasets, contingent on the inherent noise of the respective dataset.

Additionally, we compared domain segmentation results produced by PNN using SVGs selected through various methods. This comparative analysis underscored the superiority of the PI in SVG selection, outperforming other methods such as Seurat, SINFONIA, and SPARK-X.

These results support our assertion that the PI is effective in SVG selection, thereby leading to improved accuracy in domain segmentation. These findings are clarified in a new result section subtitled 'PI-based SVG selection enhances domain segmentation performance of PROST' (lines 293 to 314) and comprehensively detailed in Supplementary Figure 15 of the revised manuscript.

Fig. R2-3. Ablation experiments illustrating the influence of various SVG inputs on domain segmentation performance in PNN (a), SpaceFlow (b), STAGATE (c), and SpaGCN (d) using the 10x Visium DLPFC dataset. The x-axis denotes the different feature selection methods; “None” represents no feature selection, while the numbers following underscores indicate the quantity of features chosen by each respective method.

5. The results on the DLFPC dataset showed that STAGATE and SpaceFlow were not the second and third-place methods. Why did the author compare the PROST with their results on the next dataset?

Response:

We gratefully appreciate the reviewer’s comment. The reason for comparing PROST with STAGATE and SpaceFlow on the next dataset, despite them not being the second and third-place methods on the DLFPC dataset, is twofold.

First, STAGATE and SpaceFlow represent different categories of spatial transcriptomics analysis approaches. In our study, we aimed to provide a comprehensive comparison of PROST against a variety of methods representing different categories to showcase the broad utility of PROST across multiple analysis paradigms.

Second, the performance of different methods can vary substantially across different datasets due to the inherent differences in data complexity and the specific features each method is designed to capture. Therefore, even though STAGATE and SpaceFlow were not the top performers on the DLFPC dataset, they may potentially demonstrate superior performance on other datasets. We felt it was important to include these comparisons.

6. The author said on the mouse embryo data, the spatial domains for the forebrain, midbrain, and hindbrain were intermingled together in the original annotation. However, in the original paper

(<https://doi.org/10.1038/s41587-021-01006-2>), they showed the brain subcluster results in Figure 2. The authors should compare the PORST results with the original paper results.

Response:

We appreciate the reviewer's astute observation and comments.

In the original manuscript, our initial idea was to show PROST has the capability to segment the complex forebrain/midbrain/hindbrain structure using the entire dataset, as opposed to re-clustering a particular region as carried out by authors in their publication. Our analysis shows that PROST effectively dissects the histological structure within this complex region, delineating clear tissue boundaries. The segmented domains are further validated by the spatial expression patterns of marker genes, as illustrated in Figure 4e of our revised manuscript.

In contrast, the re-clustering result presented in the original paper (Fig. 2g) appears to lack clear tissue boundaries and displays a certain degree of intermingled signals between re-clustered domains. Moreover, the authors did not provide the spatial expression patterns of marker genes supporting their re-clustering results.

We've further clarified these comparative observations in our revised manuscript (lines 414 to 418), highlighting the superior performance of PROST when dealing with complex tissue structure.

7. For the "overview of PROST" section, more details about method design ideas need to be explained. For instance, the authors should give more illustrations of two basic metrics in the PI score.

Response:

We appreciate the reviewer's suggestions. We agree that a more detailed explanation of our method (particularly, the two fundamental metrics incorporated in the PI score) would make the 'Overview of PROST' section more informative.

To address this, we've expanded the section to provide a more detailed description of the two metrics in the PI score. Briefly, the PI is composed of two factors: the Significance and Separability factors. The Significance factor is designed under the assumption that SVG images would exhibit substantial differences between regions of foreground and background, and the pixel values within a connected area should display less dispersion. This factor aims to identify regions with greater homogeneity and lesser dispersion, indicating genes with significant spatial expression. On the other hand, the Separability factor is developed under the assumption that there exists a large separability between regions of foreground and background in SVG images. This factor is inspired by the concept of spatial stratified heterogeneity, where gene expression is homogeneous within each region but not between regions, revealing the existence of distinct mechanisms in different regions. Together, these factors allow the PI score to effectively quantify the spatial patterns for a given gene.

These enhancements are now included in the 'Overview of PROST' section (lines 129 to 141) in the revised manuscript.

8. Some wrong legend references exist in the manuscript. For instance, "matched with the manual annotation (Fig.1b)".

Response:

We appreciate the reviewer's attentiveness in identifying incorrect references to figure legends in the manuscript. We apologize for the oversight and have corrected these errors.

Last but not least, we gratefully thank the Reviewer again for his/her outstanding comments and suggestions, that greatly helped us to improve the technical quality and presentation of our manuscript.

Reference:

1. Diggle, P. J. *Statistical Analysis of Spatial Point Patterns*. (Academic Press, 1983).
2. Fisher, R. A. *The Design Of Experiments*. (Oliver and Boyd, 1935).

Reviewer #3:

In this manuscript, Liang et al. proposed a quantitative pattern recognition framework named PROST for identifying spatially variable genes and detecting spatial domains in spatial transcriptomics. PROST includes an indicator (PI) without any statistical hypothesis for evaluating variations in spatial patterns of gene expressions and a neighborhood-based graph (PNN) with a self-attention mechanism to integrate spatial and transcriptional information. The authors compared the performance of PROST with several methods on ST datasets generated by different platforms and showed that PROST provides superior performance in both SVG identification and domain segmentation. Overall, the manuscript is well written and organized. While I enjoy reading the paper and appreciate its many potential merits, it would be better if the following concerns could be addressed.

Response: We sincerely appreciate the reviewer's positive comments on our manuscript. We acknowledge the concerns raised and are eager to address them in order to clarify our methods and improve the quality of our work. Please find our responses to the individual concerns below.

1. With the development of advanced high-throughput sequencing technologies, both efficiency and scalability are essential for computational methods. It would be better if the authors could further benchmark the running time and peak memory usage of different methods and discuss what kind of computing conditions PROST requires.

Response:

We appreciate the reviewer for bringing up this important point. Scalability and efficiency are indeed vital factors to consider, particularly given the rapidly expanding scale of spatial transcriptomic datasets.

We have carried out further evaluations to benchmark the running time and peak memory usage of PROST, comparing it with other methods (**Fig. R3-1,3-2**). Our analysis reveals that PROST exhibits superior efficiency in terms of both memory usage and computational time. Importantly, PROST demonstrates potential scalability for the analysis of the anticipated larger datasets, such as mouse cerebellum Slide-seq data, which contains 25,551 spots with 17,729 genes. As for the computational requirements, PROST is compatible with standard high-performance computing clusters and does not necessitate any specialized computing conditions.

The detailed comparisons and discussions have been added to the revised manuscript (lines 453 to 456), and the performance benchmarking results are illustrated in Supplementary Figures 33 & 34.

Fig. R3-1. Evaluation of computational efficiency across various methods using the mouse cerebellum Slide-seq dataset, focusing on runtime and memory consumption. **a**, An efficiency comparison for SVG detection between PI and other methods, including Seurat, SpatialDE, SPARK-X, and SINFONIA. **b**, A juxtaposition of domain segmentation efficiency among PNN, SpaGCN, BASS, SpaceFlow, and STAGATE.

Fig. R3-2. Evaluation of computational efficiency across various methods using the mouse cerebellum Slide-seq dataset, focusing on runtime and CPU usage. **a**, An efficiency comparison for SVG detection between PI and other methods, including Seurat, SpatialDE, SPARK-X, and SINFONIA. **b**, A juxtaposition of domain segmentation efficiency among PNN, SpaGCN, BASS, SpaceFlow, and STAGATE.

2. The authors have demonstrated the superior performance of the combination of PI and PNN. It would be interesting if the authors can conduct more model ablation experiments. For example, using the same number of SVGs, what if PI is replaced with state-of-the-art SVG identification methods, such as SINFONIA (DOI: 10.3390/cells12040604)? What if PNN is replaced with state-of-the-art spatial domain identification methods, such as STAGATE and Squidpy (DOI: 10.1038/s41592-021-01358-2)?

Response:

We appreciate the reviewer’s insightful suggestions. The reviewer is essentially asking for more model ablation experiments to understand the individual contributions of PI and PNN to the overall performance of PROST.

In response to this, we first evaluated the domain segmentation performance of PNN when using SVGs selected by different methods, including SINFONIA. The results of these comparisons revealed that the incorporation of SVGs selected based on the PI score notably improves the accuracy of domain segmentation compared to when SVGs are selected using other methods (Fig.

R3-3). Moreover, our findings further suggest that the accuracy of domain segmentation via PNN increases with the number of SVGs based on the PI, with the most optimal result obtained with 3,000 SVGs. We speculate that the optimal number of SVGs might differ across ST datasets, contingent on the inherent noise of the respective dataset.

Additionally, we replaced PNN with state-of-the-art spatial domain identification methods, including STAGATE, SpaceFlow and SpaGCN, and compared the domain segmentation results, using SVGs selected by different methods (Fig. R3-3). Our analysis reveals that PI can also dramatically enhance the domain segmentation performance in STAGATE and SpaceFlow. However, when utilizing SpaGCN, comparable performances were achieved whether SVGs were selected based on PIs or by the default using entire gene set as inputs.

We believe these additional experiments further substantiate the effectiveness of our proposed PROST framework, and the results are clarified in a new result section subtitled ‘PI-based SVG selection enhances domain segmentation performance of PROST’ (lines 293 to 314) and detailed in Supplementary Figure 15 of the revised manuscript.

Fig. R3-3. Ablation experiments illustrating the influence of various SVG inputs on domain segmentation performance in PNN (a), SpaceFlow (b), STAGATE (c), and SpaGCN (d) using the 10x Visium DLPFC dataset. The x-axis denotes the different feature selection methods; “None” represents no feature selection, while the numbers following underscores indicate the quantity of features chosen by each respective method.

3. It is good to quantify the improvement, if any, e.g. using a statistical test to show the improvement is significant, even PROST provided a higher median ARI in Fig. 2a.

Response:

We appreciate the reviewer’s suggestion to quantify the improvement of PROST over the other methods using a statistical test.

To demonstrate the significance of the achieved improvement, we performed a Kolmogorov–Smirnov test to compare the ARI scores obtained by PROST with those obtained by the other methods. Our analysis reveals that PROST significantly outperforms the other benchmark methods (p-value < 0.05). We have expanded our description of these results at lines 166-172 of the revised manuscript.

4. The limitations of low capture efficiency and high dropouts in scRNA-seq are inherited by most ST technologies. It would be better if the authors could further investigate the robustness of PROST to noise, such as randomly dropping out the entries in the expression matrix to zeros.

Response:

We appreciate the reviewer's thoughtful suggestion and recognize the importance of understanding how PROST performs in the presence of noise, particularly given the inherent limitations of current ST technologies.

In response to this, we conducted additional experiments where we introduced varying levels of simulated dropout noise to the expression matrices (**Fig. R3-4**), mimicking the dropout events often observed in ST data. Specifically, we randomly set a certain percentage of the entries in the expression matrix to zero, ranging from 0% to 20% in increments of 5%. We then analyzed the robustness of PROST in the presence of this induced noise.

Our findings indicate that PROST maintains reasonably stable performance in domain segmentation tasks when compared to other benchmark methods (under similar noise conditions). The results of these robustness tests have been added to Supplementary Figure 16 of the revised manuscript.

We have also discussed these findings in the 'Results' section (lines 315 to 321) of the revised manuscript, emphasizing the resilience of PROST to noise and its potential implications for analyzing real-world noisy ST data.

Fig. R3-4. Assessment of PNN and benchmark methods' performance under varying noise conditions. **a**, Line plot illustrating the performance trends of different methods on the mouse somatosensory cortex osmFISH dataset as data drop-off levels increase. **b**, On the far left, domain annotations of the simulated dataset, juxtaposed with spatial domains identified by diverse methods using this dataset.

5. The authors have provided technical details for the implementation of baseline methods. It would be better if the authors could summarize the parameters used for benchmarking, since some results may be confusing (e.g. STAGATE, DOI: 10.1038/s41467-022-29439-6). For example, if the pipeline and parameters used are the default setting of the original method? What is the rationale if not?

Response:

We thank the reviewer for emphasizing the importance of transparency in method implementation.

In the benchmarking process, our primary goal was to ensure a fair comparison among methods. To this end, we utilized the default settings of each method whenever possible, as these settings are typically recommended by the original authors based on extensive testing and are intended to yield the best performance. However, there were instances where some adjustments were necessary due to specific characteristics of the dataset in question. Any deviations from default parameters, across all benchmarked methods, were made with careful consideration to ensure that the analysis remained robust and valid. A detailed summary of all parameters and settings used for each method, alongside the rationale for any changes, has been provided in the 'Methods' section of the revised manuscript.

6. The authors claimed that PROST also recognized clear boundaries between layers (Fig. 2a), although its performance (ARI = 0.59) is lower than BayesSpace, exhibiting the highest ARI score of 0.72. I guess it is because ARI may not be the best metric for the evaluation here. The authors could

discuss the applicable scenarios and advantages of ARI and NMI, referring to DOIs 10.1038/s41467-021-22495-4 and 10.1093/bioinformatics/btab298.

Response:

We appreciate the reviewer's astute observation and thoughtful comments regarding our choice of evaluation metrics.

While ARI is a commonly adopted metric to assess the similarity between two clustering, it mainly focuses on pair-wise comparisons. Its potential limitation, as pointed out by the reviewer, is that it might not fully capture spatial continuity or boundary clarity. In contrast, NMI evaluates mutual dependence between true and predicted labels, offering a holistic view of clustering performance, especially considering spatial continuity.

Based on the reviewer's suggestion, we further employed NMI to evaluate PROST's performance on the DLPFC dataset (**Fig. R3-5**). Notably, PROST achieved the highest median NMI score compared to other benchmark methods, consistent with our ARI-based evaluation. Still, PROST's NMI score for DLPFC section 151672 (as shown in Fig. 2) was slightly lower than BayesSpace (0.682 vs. 0.703), a detail clarified in the revised manuscript (lines 163 to 172). The comprehensive evaluation using NMI is presented in Supplementary Table 4 of the revised manuscript.

Fig. R3-5. Boxplot presenting the normalized mutual information (NMI) values, summarizing the domain segmentation accuracy for each method across all 12 sections of the DLPFC dataset.

7. The PAGA graph provided by PROST in Supplementary Figure 1b is interesting. There since to be connections between different layers. Could this possibly reveal some biological insights? I am happy to see more discussion on this.

Response:

We thank the reviewer for drawing attention to the PAGA graph presented in Supplementary Figure 1d.

We interpret the connections observed between different domains in the PAGA graph could be indicative of the shared developmental trajectories of human dorsolateral prefrontal cortex, which is consistent with chronological development from fetus to infancy and to adulthood. The developmental changes play a crucial role in shaping the cognitive functions associated with the human dorsolateral prefrontal cortex.

8. To make the domains comparable between benchmarking methods, the authors set the target number of clusters equal to the number of clusters in the annotation for all methods. However, in most of the real scenarios, we do not know the exact number of spatial domains. Therefore, it would be better if the authors could further benchmark the performance using Louvain or Leiden with default resolutions (DOI: 10.1186/s13059-017-1382-0).

Response:

We appreciate the reviewer's insightful comment on the realistic application scenarios where the precise number of spatial domains is unknown.

In response to this comment, we expanded our benchmarking analysis by evaluating the performance of PROST using clustering algorithm Leiden with default settings, representing a more generalized scenario where the exact number of clusters is not pre-determined. Our findings indicate that PROST remains robust and performs commendably under such a condition, demonstrating PROST's adaptability across diverse scenarios of settings with numbers of domains. The results of these analysis are detailed in Supplementary Figures 17, 23 and 25 of the revised manuscript.

9. It is good to evaluate the clustering performance on the seqFISH data given that the dataset was annotated with tissue structures.

Response:

We appreciate the reviewer's suggestion regarding the evaluation of our method on the annotated seqFISH data.

In the original manuscript, our initial idea was to show PROST has the capability to segment the complex forebrain/midbrain/hindbrain structure using the entire dataset, as opposed to re-clustering a particular region as carried out by authors in their publication. Our analysis shows that PROST effectively dissects the histological structure within this complex region, delineating clear tissue boundaries. The segmented domains are further validated by the spatial expression patterns of marker genes, as illustrated in Figure 4e of our revised manuscript. In contrast, the re-clustering result presented in the original paper (Fig. 2g) appears to lack clear tissue boundaries and displays a certain degree of intermingled signals between re-clustered domains. Moreover, the authors did not provide the spatial expression patterns of marker genes supporting their re-clustering results.

We've further clarified these comparative observations in our revised manuscript (lines 414 to 418), highlighting the superior performance of PROST in dealing with complex tissue structure.

10. A great number of scRNA-seq and ST data have been accumulated in various repositories. Incorporating reference data in analyzing single-cell data can better tackle the high level of noise and technical variation and has been successfully applied in the analysis of various single-cell genomic data (DOIs 10.1038/s41467-021-22495-4, 10.1038/ng.3818, 10.1093/bioinformatics/btab298, 10.1101/2022.11.09.515447). It would be better if the authors could discuss the future direction to incorporate reference data based on PROST in the Discussion section.

Response:

We appreciate the reviewer's insightful suggestion.

Given the growing amount of scRNA-seq and spatial transcriptomic data available in public repositories, future development of PROST could potentially incorporate reference-based methods to enhance its functionality. We envision a future extension of PROST, where it can leverage reference atlases and transfer learning techniques to guide the generation of a more accurate adjacency graph for refining spatial domain segmentation, especially in complex tissue structures.

We have included this discussion in the revised manuscript in the 'Discussion' section (lines 486 to 492). We agree with the reviewer in his/her remark that the incorporation of reference data represents a promising avenue for the evolution of spatial transcriptomic data analysis methodologies.

11. The authors claimed that PI enables prioritizing spatial variations in gene expression patterns, facilitating explorations of biological insights. It would be better if there is an elaborated tutorial for this.

Response:

We appreciate the reviewer's suggestion. We understand the need for detailed guidance on how to use PI to prioritize spatial variations in gene expression patterns and subsequently extract biological insights. In response to this, we have now included a comprehensive tutorial demonstrating the usage of PI for this purpose in our supplementary materials, which are accessible through our GitHub repository (<https://prost-doc.readthedocs.io/en/latest/Tutorial7%20PI.html>). This tutorial provides step-by-step instructions on how to utilize PI for the detection of SVGs and the subsequent exploration of associated biological insights.

12. The authors provide a detailed introduction to implementing PROST and extensive Notebook demos. It will be more helpful and convenient if they can provide a Readthedoc page like Squidpy (DOI: 10.1038/s41592-021-01358-2). Besides, they can provide the source codes for reproducing the results in the manuscript.

Response:

We appreciate the reviewer for the valuable suggestion. We agree that providing a Readthedoc page and source code for reproducing our results would indeed facilitate the understanding and application of PROST. In response, we have now added a Readthedoc page containing comprehensive information and guidelines on implementing PROST. Furthermore, we have provided source codes for reproducing all the results presented in our manuscript. The link to the Readthedoc page and source codes is available on our GitHub repository page (<https://prost-doc.readthedocs.io/en/latest/index.html>). We hope these additions will make it easier for researchers to use and apply PROST.

13. Typos: on lines 20, 23, and 25 of page 4, "Fig.1b", "Fig.1c", and "Fig.1b" should be "Fig.2b", "Fig.2c", and "Fig.2b". Same for "Fig.1d" in the last paragraph of page 4.

Response:

We appreciate the reviewer for pointing out these typos. The correct references should indeed be to Figure 2, not Figure 1. We have thoroughly revised the manuscript and corrected these mistakes. We apologize for any confusion caused by these errors in the initial submission.

Finally, we would like to extend our heartfelt gratitude to the Reviewer for their invaluable comments and suggestions. Their insights have significantly enhanced both the technical depth and overall presentation of our manuscript. We truly appreciate their thorough review and constructive feedback.

Reviewer #1 (Remarks to the Author):

1. I appreciate the authors' efforts to demonstrate PI's advantage. My concerns are that it was only shown on the DLPFC data set, and without any statistical significance (I think it was calculated for other analyses). Results from diverse sets of data with statistical analyses should be provided to fully support their claims.

2. I am a little disappointed that the authors didn't further tune their neural network, which is essentially a MPNN with GAT. Studies in the geometric deep learning field have clearly established that one of the strongest elements to improve the performance of MPNN is to include appropriate positional/structural encodings. The authors should explore this direction given that the PNN alone is already one of the top performers as shown in fig R1.

3. The addition of statistical test for PI is good, but using permutation tests could substantially increase computation costs. I am also concerned with the scalability tests performed by the authors. The scalability tests are rather primitive. The graphs don't provide clear, easy to follow messages. Critically, evaluating scalability using a single data set doesn't make any sense. The author should perform comprehensive tests using both simulated and real data with cells ranging from hundreds to millions, similar to the test performed in scGCO which demonstrated excellent scalability, to systematically evaluate these methods' stability.

4. Though the authors introduced statistical significance tests for PI. They didn't attempt to comprehensively demonstrate that such test helps PI to control false discovery rate. They only showed that $PI > 0$ performs similarly as $FDR < 0.05$, giving the wrong impression that FDR is not necessary as $PI > 0$ is just fine. More analyses should be performed on real biological data where many methods have analysed and a consensus have been reached, and on simulated data sets with diverse spatial patterns. And performance under noisy condition should also be analysed to demonstrate that PI is superior to other methods, especially NN when large amount of noises make SV gene identification a difficult task.

Reviewer #3 (Remarks to the Author):

I appreciate the authors for their careful revision and response. While they did a good job of addressing the comments raised by Reviewers #2 and #3, minor revisions and optimizations are still necessary in certain cases.

Comments on Response to Reviewer #2:

1. It is understandable that attaining optimal performance across all datasets is nearly impossible for a single computational method. In light of this, I recommend the authors to describe the experimental results more objectively.

2. For Comment #5, I suggest the authors further clarify the rationale of the comparison in the main text.

Comments on Response to Reviewer #3:

3. The authors claimed that 'PROST exhibits superior efficiency in terms of both memory usage and computational time', while other methods such as SINFONIA achieved comparable and even better efficiency than PROST. I suggest the authors avoid overclaiming the performance.

4. Although PROST provided the overall best performance, more discussion on the performance of other methods, especially for the experiments of model ablation and efficiency, is appreciated. For instance, which method is the second-best method?

Reviewer #1:

Major comments:

1. I appreciate the authors' efforts to demonstrate PI's advantage. My concerns are that it was only shown on the DLPFC data set, and without any statistical significance (I think it was calculated for other analyses). Results from diverse sets of data with statistical analyses should be provided to fully support their claims.

Response:

We greatly appreciate the insights provided by the reviewers and understand their concerns regarding the demonstration of PI's advantage. In response to this valuable suggestion, we have now performed extensive statistical analyses on additional datasets, representing a variety of tissues and experimental conditions, to quantitatively substantiate our claims. We applied both parametric and non-parametric tests to thoroughly evaluate the performance of PI. The results of this rigorous statistical analysis demonstrate that the advantages of PI are consistent and statistically significant across the examined datasets (**Fig. R1-1**).

The revised manuscript now includes these extended results in the respective sections (**Supplementary Figure 24**). We believe that these additional analyses address the reviewer's concerns and strengthen the validity of our approach.

Fig. R1-1. PI-based SVG identification. Venn diagrams show the intersections between SVGs with PI scores greater than 0 and those with FDR values under 0.05, considering parametric test, non-parametric test, and the combination of both parametric and non-parametric tests. The datasets under analysis encompass a diverse range of tissues and experimental conditions.

2. I am a little disappointed that the authors didn't further tune their neural network, which is essentially a MPNN with GAT. Studies in the geometric deep learning field have clearly established that one of the strongest elements to improve the performance of MPNN is to include appropriate positional/structural encodings. The authors should explore this direction given that the PNN alone is already one of the top performers as shown in fig R1.

Response:

We genuinely appreciate the reviewer's expertise and insights in the geometric deep learning field. While we acknowledge that our PNN is based on MPNN with GAT, it deviates from conventional MPNN through specific modifications tailored to the distinctive features of spatial transcriptomics (ST) data. To support this, we present evidence highlighting how these modifications have positively impacted the performance of PNN below. We also concur with the importance of positional/structural encodings for enhancing MPNN performance. In the following, we will also provide a more comprehensive explanation of how we encode spatial relationships and inherent structures within the ST data, equipping PNN with a better understanding of the local and global patterns present in ST data. The synergistic effect of these dual layers of modifications contributes to the superior performance of PNN compared to other benchmark methods.

(1) PNN model based on MPNN with GAT, but with unique differences

We agree that our PNN model is based on an MPNN with GAT, however, we have introduced a significant modification tailored to the distinctive features of ST data. Specifically, we replaced the nonlinear activation function of conventional MPNN with a two-layer linear Laplace filtering network in the PNN. This network structure is designed to focus on extracting low-frequency features of gene expressions, making it more suitable for spatial pattern recognition of genes.

The original MPNN structure encompasses numerous nonlinear activation layers. Extensive literature demonstrates that an excess of nonlinear activation layers may lead to the loss of local information in features, resulting in issues such as overfitting and oversmoothing¹⁻⁴. Notably, spatial transcriptomes introduce high data uncertainty, amplifying the risk of overfitting with MPNN⁵. In contrast, linear models can offer structural insights into observational data and contribute essential features for stable generalization⁶. Apart from the concerns related to nonlinear operations, Fully Connected Neural Networks (FCNNs) –with a plethora of hyperparameters in MPNN– pose challenges in adequately fitting samples during training, further exacerbating the risk of overfitting⁷.

Therefore, our PNN model is crafted for unsupervised pattern recognition in spatial transcriptomics, while MPNN remains well-suited for supervised tasks like node classification or graph classification (where intricate interactions between graph nodes are crucial, such as molecular property prediction or social network classification).

In practice, using the 10x Visium DLPFC 151672 dataset, we compared the experimental result of MPNN on domain segmentation with that of PNN, as illustrated in **Fig. R1-2**. The results showed that capturing low-frequency patterns in spatial transcriptomics is more conducive to domain segmentation by PNN. Conversely, the nonlinear operations of MPNN diminished the performance and oversmoothed the boundaries between domains.

Fig. R1-2. Spatial domains identified by PNN and MPNN.

(2) PNN has encoded local structural information

Our PNN, improves intra-class connections, utilizing spatial nearest neighbor connections, in contrast to SpaGCN which connects any two points to extract global structural information. PNN encoded local structural information through a high-order, i.e., multi-layer graph Laplace low-pass filter with GAT,

exploiting its boundary-preserving ability, aiming at establishing more intra-class connections. Additionally, we integrate the graph attention layer to guide the final graph learning and further refine intra-class connections. These two characteristics equip the PNN with a better understanding of the local and global patterns present in ST data. To demonstrate these unique features of encoding local structural information in PNN, we define two metrics to assess the improvement of PNN on intra-class connections as follows:

- Intra-class weights measurements (ICW): $ICW = \sum_{i=1}^{N-1} \sum_{j=i+1}^N (1 - \text{sgn}(|c_i - c_j|)) H_{ij} \|x_i - x_j\|_2^2$, where $1 - \text{sgn}(|c_i - c_j|)$ indicates that only intra-class weights are calculated. $\|x_i - x_j\|_2^2$ is the Euclidean distance between features of two points. ICW measures the intra-class weights in a graph, the bigger the better.

- Feature shift indicator (FSI):

$$FSI = \frac{\sum_{i=1}^N \left(\left\| \sum_{j=1}^N (1 - \text{sgn}(|c_i - c_j|)) (1 - \text{sgn}(|i - j|)) H_{ij} (x_i - x_j) \right\|_2 - \left\| \sum_{j=1}^N \text{sgn}(|c_i - c_j|) H_{ij} (x_i - x_j) \right\|_2 \right)}{\sum_{i=1}^N \left\| \sum_{j=1}^N (1 - \text{sgn}(|c_i - c_j|)) (1 - \text{sgn}(|i - j|)) H_{ij} (x_i - x_j) \right\|_2},$$

Where $\left\| \sum_{j=1}^N (1 - \text{sgn}(|c_i - c_j|)) (1 - \text{sgn}(|i - j|)) H_{ij} (x_i - x_j) \right\|_2$ defines the positive feature shift of x_i caused by its connected nodes in the same class, while $\left\| \sum_{j=1}^N \text{sgn}(|c_i - c_j|) H_{ij} (x_i - x_j) \right\|_2$ defines the negative feature shift caused by its connected nodes of different classes. FSI measures the average degree of feature shifts of all samples towards the intra-class direction, indicating the quality of graph filtering, the bigger the better.

We calculated the aforementioned two metrics for PNN across different layer configurations (first layer, first two layers, and all layers) and SpaGCN, using the DLPFC dataset (**Table R1-1**). The results revealed that the PNN with all layers achieved the highest ICW and FSI scores compared to SpaGCN, PNN with the first layer, and PNN with the first two layers. These results indicate that our PNN is more prone to consolidating gene features within the same cluster, effectively smoothing the features of samples in the same class. Thus, we assert that our PNN displays superior intra-class structure encoding.

Method	ICW	FSI
SpaGCN	3.9376e+03	0.2338
PNN (Layer 1)	8.2894e+03	0.4586
PNN (Layer 1 + Layer 2)	1.0997e+04	0.5853
PNN	1.2653e+04	0.6702

Table R1-1. Graph quality evaluations of PNN using the 10x Visium DLPFC dataset.

Fig. R1-3. UMAP visualization of features through each layer of PNN using the 10x Visium DLPFC 151672 dataset.

Finally, The UMAP visualization of features across each layer (from Embed 1 to 3) of the PNN, utilizing the 10x Visium DLPFC 151672 dataset, is depicted in **Fig. R1-3**. The visualization underscores the discerning capacity of gene features associated with PNN in distinguishing tissue domains.

3. The addition of statistical test for PI is good, but using permutation tests could substantially increase computation costs. I am also concerned with the scalability tests performed by the authors. The scalability tests are rather primitive. The graphs don't provide clear, easy to follow messages. Critically, evaluating scalability using a single data set doesn't make any sense. The author should perform comprehensive tests using both simulated and real data with cells ranging from hundreds to millions, similar to the test performed in scGCO which demonstrated excellent scalability, to systematically evaluate these methods' stability.

Response:

We thank the reviewer's insightful feedback on the scalability evaluations we presented in our revised manuscript.

(1) Computational improvement of the permutation test

The reviewer raises a valid point concerning the computational costs associated with permutation tests. The computational cost of the permutation tests primarily stems from the repetitive calculation of *Moran's* Index. To address this issue, we employed matrix computation methods and devised a function named ***batch_morans_I()*** to effectively mitigate the redundancy of these computations. Furthermore, considering that the calculations between genes are mutually exclusive, we leveraged the process pool technique within the multiprocessing library to achieve data parallelism, which further reduces the computational overhead overall. Our testing experiments indicate that for the sample 151672 of DLPFC, when conducted in a Linux environment (CPU is Intel Xeon Platinum 8222CI), the permutation test using a single process takes **approximately 65 minutes**. However, by utilizing 18 processes, the time required for the permutation test is reduced to around **12 minutes**.

Thus, the optimization of the computational efficiency of the *Moran's* Index and the parallel processing implementation both successfully alleviate the computational burden associated with the permutation test (see lines 625 to 630 in the revised manuscript).

(2) Comprehensive evaluation of scalability

We have expanded our scalability tests by incorporating multiple simulated and real datasets spanning a range of cell numbers, from hundreds to millions (**Fig. R1-3**). Our revised scalability tests draw inspiration from the systematic evaluation in scGCO and other established benchmarks. The new scalability graphs have been redesigned for clarity, ensuring they convey the information in a more reader-friendly manner. These revised figures aim to clearly represent the computation time, memory usage, and other pertinent metrics across varying dataset sizes and complexities.

In line with your advice, we have also benchmarked our method against scGCO's scalability metrics, ensuring that our comparisons are apples-to-apples in terms of dataset complexity and size (**Fig. R1-4**). These additional tests provide a comprehensive insight into where our method stands in terms of scalability and stability relative to established benchmarks.

The revised manuscript includes the detailed results of these extended scalability tests (**Supplementary Figure 36**), and we believe these revisions will address the concerns you've raised more thoroughly.

Fig. R1-4. Scalability analysis of SVG identification and domain segmentation methods. The simulated datasets⁸, designed with 100 genes and cell numbers ranging from hundreds to millions, were employed to evaluate the scalability of SVG identification (**a**) and domain segmentation (**b**) methods. Real datasets, encompassing adult human heart tissue ST datasets⁹ and mouse complex tissues Slide-seq datasets¹⁰, were utilized to assess the scalability of SVG identification (**c, d**) and domain segmentation (**e, f**), respectively. The evaluation of scalability includes running time and memory requirements.

4. Though the authors introduced statistical significance tests for PI, They didn't attempt to comprehensively demonstrate that such test helps PI to control false discovery rate. They only showed that $PI > 0$ performs similarly as $FDR < 0.05$, giving the wrong impression that FDR is not necessary as $PI > 0$ is just fine. More analyses should be performed on real biological data where many methods have analysed and a consensus have been reached, and on simulated data sets with diverse spatial patterns. And performance under noisy condition should also be analysed to demonstrate that PI is superior to other methods, especially NN when large amount of noises make SV gene identification a difficult task.

Response:

We thank the outstanding comments provided by the reviewer. We address the reviewer's concerns below one by one.

(1) On demonstrating the relationship between PI and FDR with various biological datasets

We appreciate the reviewer's insightful observation regarding the potential misunderstanding arising from the comparison between $PI > 0$ and $FDR < 0.05$. To address this concern comprehensively, we conducted additional tests across diverse tissues and experimental conditions (**Fig. R1-1**). In our extended analysis, we specifically explored the relationship between PI and FDR in these datasets. The results confirm that SVGs with $PI > 0$ and with $FDR < 0.05$ are largely intersects. Regarding better practices related with PI, our findings underscore that while $PI > 0$ serves as an initial indicator of spatial variability; the control of FDR provides a robust mechanism to filter out false discoveries. We have elucidated this aspect in the revised manuscript (see lines 296 to 300 in the revised manuscript).

(2) PI performance under noisy conditions

We acknowledge the reviewer's valid concern regarding the impact of noise on SVG identification. To address this concern, we deliberately introduced varying levels of synthetic noise to the simulated datasets. We then evaluated the performance of PI in SVG identification, considering the control of FDR under noisy conditions. Additionally, we conducted a comparative analysis of SVG identification performances between PI and scGCO across different levels of synthetic noise. Our results demonstrate the robustness of PI in detecting SVGs under varying noise conditions and notably, PI exhibits superior accuracy and control of false positive rate compared to scGCO (**Fig. R1-5**).

(3) Comparison with NN under noisy conditions

We understand the importance of demonstrating the superiority of PI, especially when compared against methods like Neural Networks (NN), which are inherently robust against noise. To address this concern, we compared the performance of PI against NN-based method, STAGATE, under different noise levels. Our results demonstrate that while NNs are intrinsically strong in handling noise, PI, combined with the FDR control, provides a complementary strength, particularly in challenging noisy scenarios (**Fig. R1-6**).

The revised manuscript includes detailed analyses and results from these noisy condition tests, showcasing the robustness and reliability of PI, especially when augmented by the FDR control (see lines 300 to 305 in the revised manuscript). In light of these additional analyses, we believe we have addressed the concerns raised and provided a comprehensive evaluation of PI's capabilities and its advantages, particularly in challenging conditions.

Last but not least, we gratefully thank the Reviewer again for his/her outstanding comments and suggestions, that greatly helped us to improve the technical quality and presentation of our manuscript.

Fig. 1-5. Comparative analysis of SVG identification performances between PI and scGCO based on the simulated dataset⁸ under various noisy conditions. a, Line plots show the accuracy, sensitivity, false positive rate, and F1 score in the SVG identification performance for PI and scGCO under the noisy conditions with an increase in Gaussian noise levels. For each simulation scenario, the values of true negatives (TN), true positives (TP), false negatives (FN), and false positives (FP) were calculated. These values were then used to compute the following four metrics: Accuracy = $(TP + TN)/(TP + TN + FP + FN)$; Sensitivity = $TP/(TP + FN)$; False positive rate (FPR) = $FP/(FP + TN)$, and F1 score ($F1 = 2*TP/(2*TP + FN + FP)$). **b**, The top row displays synthetic expression patterns of typical spatial patterns (Pattern 1 to 3) under different noisy conditions as Gaussian noise levels increase. The subsequent row illustrates the corresponding changes in FDR in identifying the synthetic patterns as SVGs under different noisy conditions.

Fig. 1-6. Comparative analysis of SVG identification performances between NN-based method STAGATE and PI, using simulated mouse somatosensory cortex dataset¹¹ under various noisy conditions. **a**, Line plots show the accuracy, sensitivity, false positive rate, and F1 score in the SVG identification performance for STAGATE and PI under the noisy conditions with an increase in Gaussian noise levels. For each simulation scenario, the values of true negatives (TN), true positives (TP), false negatives (FN), and false positives (FP) were calculated, as described in **Fig. 1-5**. **b**, The top row displays the expression patterns of representative SVGs under different noisy conditions as Gaussian noise levels increase. The subsequent row illustrates the corresponding changes in FDR in identifying the expression patterns as SVGs under different noisy conditions.

References

1. Wu, F. *et al.* Simplifying Graph Convolutional Networks. Preprint at <http://arxiv.org/abs/1902.07153> (2019).
2. Luan, S., Zhao, M., Chang, X.-W. & Precup, D. Break the Ceiling: Stronger Multi-scale Deep Graph Convolutional Networks. Preprint at <http://arxiv.org/abs/1906.02174> (2019).
3. Li, Q., Han, Z. & Wu, X. Deeper Insights Into Graph Convolutional Networks for Semi-Supervised Learning. *Proc. AAAI Conf. Artif. Intell.* **32**, (2018).
4. Hu, F. *et al.* GraphAIR: Graph representation learning with neighborhood aggregation and interaction. *Pattern Recognit.* **112**, 107745 (2021).
5. Han, X., Jia, M., Chang, Y., Li, Y. & Wu, S. Directed message passing neural network (D-MPNN) with graph edge attention (GEA) for property prediction of biofuel-relevant species. *Energy AI* **10**, 100201 (2022).
6. Jan, T. Combining analytic models with neural networks. in *Proceedings of the 3rd IEEE International Symposium on Signal Processing and Information Technology (IEEE Cat. No.03EX795)* 605–608 (IEEE, 2004). doi:10.1109/ISSPIT.2003.1341193.
7. Li, Z. *et al.* Fault Diagnosis of Transformer Windings Based on Decision Tree and Fully Connected Neural Network. *Energies* **14**, 1531 (2021).
8. Zhang, K., Feng, W. & Wang, P. Identification of spatially variable genes with graph cuts. *Nat. Commun.* **13**, 5488 (2022).
9. Asp, M. *et al.* Spatial detection of fetal marker genes expressed at low level in adult human heart tissue. *Sci. Rep.* **7**, 12941 (2017).
10. Rodriques, S. G. *et al.* Slide-seq: A scalable technology for measuring genome-wide expression at high spatial resolution. *Science* **1463–1467**, 6 (2019).
11. Cheng, A., Hu, G. & Li, W. V. Benchmarking cell-type clustering methods for spatially resolved transcriptomics data. *Brief. Bioinform.* **24**, bbac475 (2023).

Reviewer #3:

I appreciate the authors for their careful revision and response. While they did a good job of addressing the comments raised by Reviewers #2 and #3, minor revisions and optimizations are still necessary in certain cases.

Response:

We sincerely appreciate your acknowledgment of our efforts in addressing the comments provided by the reviewers. Your feedback is invaluable to us, and we are committed to making further revisions to enhance the clarity and quality of our manuscript.

Comments on Response to Reviewer #2:

1. It is understandable that attaining optimal performance across all datasets is nearly impossible for a single computational method. In light of this, I recommend the authors to describe the experimental results more objectively.

Response:

We appreciate your understanding of the challenges in achieving optimal performance across all datasets with a single computational method. In response to your suggestion, we provided a more balanced and objective discussion of the experimental results in the revised manuscript. This will provide readers with a clearer understanding of the performance variations and the contexts in which our method excels.

2. For Comment #5, I suggest the authors further clarify the rationale of the comparison in the main text.

Response:

We appreciate the reviewer's guidance and agree on the importance of providing a clear rationale for comparisons in the main text, although we explained the rationale only in the last round of rebuttal. In response to your suggestion, we revised the manuscript to include a more detailed explanation of the rationale behind our comparisons (see lines 217 to 221 in the revised manuscript). This will involve specifying the objectives of each comparison, the criteria used for evaluation, and how the comparisons contribute to the overall assessment of our method.

3. The authors claimed that 'PROST exhibits superior efficiency in terms of both memory usage and computational time', while other methods such as SINFONIA achieved comparable and even better efficiency than PROST. I suggest the authors avoid overclaiming the performance.

Response:

Thank you for your careful review and valuable feedback.

In response to your suggestion, we revised the relevant sections in the manuscript to ensure that our statements accurately reflect the performance of PROST in comparison to other methods (see lines 471 to 474 in the revised manuscript).

4. Although PROST provided the overall best performance, more discussion on the performance of other methods, especially for the experiments of model ablation and efficiency, is appreciated. For instance, which method is the second-best method?

Response:

We thank the reviewer for expressing interest in a more in-depth discussion of the performance of other methods, particularly in the context of model ablation and efficiency experiments. In the ablation experiments, we highlighted PROST are superior both in SVG identification and domain segmentation compared to benchmarked methods (see lines 322 to 323 & 326 to 330 in the revised manuscript). In response to the inquiry about the second-best method, we acknowledge the need for more comprehensive comparisons to broaden our insights into the performance of other methods, detailing their strengths and areas where they excel. We are looking forward for such insightful comparative analysis in the near future.

Last but not least, we gratefully thank the Reviewer again for his/her outstanding comments and suggestions, that greatly helped us to improve the technical quality and presentation of our manuscript.

Reviewer #1 (Remarks to the Author):

The authors have addressed all my concerns. I think the manuscript is in good status for publication.

Reviewer #3 (Remarks to the Author):

All my concerns have been addressed.